# Scalable Language Model with Generalized Continual Learning

**Bohao PENG**[†]   **Zhuotao TIAN**[‡]   **Shu LIU**[‡]   **Mingchang YANG**[†]   **Jiaya JIA**[†]
[†] The Chinese University of Hong Kong     [‡] SMartMore

## Abstract

Continual learning has gained increasing importance as it facilitates the acquisition and refinement of scalable knowledge and skills in language models. However, existing methods typically encounter strict limitations and challenges in real-world scenarios, such as reliance on experience replay, optimization constraints, and inference task-ID. In this study, we introduce the Scalable Language Model (SLM) to overcome these limitations within a more challenging and generalized setting, representing a significant advancement toward practical applications for continual learning. Specifically, we propose the Joint Adaptive Re-Parameterization (JARe), integrated with Dynamic Task-related Knowledge Retrieval (DTKR), to enable adaptive adjustment of language models based on specific downstream tasks. This approach leverages the task distribution within the vector space, aiming to achieve a smooth and effortless continual learning process. Our method demonstrates state-of-the-art performance on diverse backbones and benchmarks, achieving effective continual learning in both full-set and few-shot scenarios with minimal forgetting. Moreover, while prior research primarily focused on a single task type such as classification, our study goes beyond, with the large language model, *i.e.*, LLaMA-2, to explore the effects across diverse domains and task types, such that a single language model can be decently scaled to broader applications. The code is available on the project website[1].

## 1 Introduction

Human-level intelligence demonstrates the remarkable ability to continuously acquire new knowledge and skills while retaining previously learned information. Although deep learning in language models has achieved significant advancements recently, it still faces challenges in retaining and accumulating knowledge when dealing with sequential tasks. It is also known as the "catastrophic forgetting" phenomenon, which refers to the potential loss of previously learned information caused by the distribution shift during the fine-tuning process for novel tasks (McCloskey & Cohen, 1989).

Despite considerable efforts to tackle the aforementioned challenges, recent studies on continual learning in language models still encounter significant limitations. Specifically, shown in Fig. 1 (a), the replay-based methods (Rebuffi et al., 2017; Romanov et al., 2018), require access to the previously learned data, leading to additional demands on resources for continual training. This approach also raises potential privacy concerns. Then, the regularization-based approaches Huang et al. (2021); Aljundi et al. (2018) (Fig. 1 (b)) exhibit vulnerability in long task sequences and struggle to strike a balance between forgetting and adaptability to specific tasks. And, certain architecture-based methods (Razdaibiedina et al., 2023) (Fig. 1 (c)) rely on task-ID during inference, which poses challenges in practical scenarios where obtaining task-IDs for individual runs may not be feasible. Besides, most previous methods have primarily focused on a single task type, such as text classification, neglecting the broader spectrum of language-related tasks (Qin & Joty, 2021). These issues deprecate the efficacy and greatly hinder the practical applications of continual learning.

In this paper, our objective is to extend the application of continual learning to a more practical and generalized setting without relying on experience replay, optimization constraints, or inference task-ID, which enables agile adaptation to novel tasks. To this end, we propose the Scalable Language

---

[1]https://github.com/Pbihao/SLM
[2]Correspondence to Zhuotao Tian(`tianzhuotao@gmail.com`).

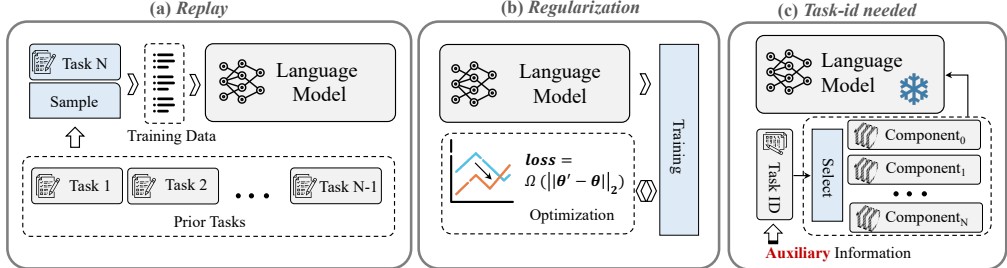

Figure 1: Illustration depicting the framework comparison of various previous methods.

Model (SLM), which *efficiently scales base language model to novel tasks in different domains without compromising the performance of the witnessed ones*.

SLM incorporates vector space retrieval into the language model, which aids in achieving scalable knowledge expansion and management, ultimately enhancing its capabilities and skill set. It comprises two primary components: Joint Adaptive Re-parameterization (JARe) and Dynamic Task-related Knowledge Retrieval (DTKR). Assuming that each task is associated with a distinct distribution in the vector space (Finn et al., 2017), the DTKR technique is utilized to identify the most relevant knowledge for each input instance. The relevant knowledge is preserved as a compilation of weight increments that leverage low-rank adaptation techniques to mitigate computational expenses (Hu et al., 2021). Then, these weight increments are employed by JARe techniques to achieve adaptive re-parameterization of the pre-trained model, with the objective of effectively aligning it with specific downstream tasks according to the task distribution.

Extensive experiments demonstrate remarkable efficacy and stability of our method on widely recognized benchmarks, reaching state-of-the-art performance on various models, including BERT, T5 and the latest LLaMA-2 (Devlin et al., 2018; Qin & Joty, 2021; Touvron et al., 2023). Our method achieves an impressive up to $80\%$ reduction in forgetting, with only a minimal $0.5\%$ performance degradation on the BERT benchmark. Unlike previous literature that primarily focuses on a single task like classification, our study pushes the boundaries by exploring continual learning across multiple task types in various domains. This comprehensive analysis highlights the superior generalization ability of our approach, making it applicable to a wider range of real-world applications.

In summary, the primary contributions of this paper can be summarized as follows:

- We propose the Scalable Language Model (SLM) as a model-agnostic solution for scalable acquisition of knowledge and skills. SLM eliminates dependencies on experience replay, optimization constraints, and inference task-IDs in a generalized continual learning setting.

- SLM incorporates vector space retrieval into the language model, with two primary components: Joint Adaptive Re-parameterization (JARe) and Dynamic Task-related Knowledge Retrieval (DTKR). Extensive experiments conducted on standard continual learning benchmarks demonstrate its remarkable superiority over previous state-of-the-art methods.

- Our study goes beyond previous literature by exploring continual learning across multiple task types from diverse domains, showcasing the superior generalization ability.

## 2 PRELIMINARIES

**Continual learning** aims to facilitate ongoing knowledge acquisition from sequential tasks while mitigating the issue of catastrophic forgetting. Specifically, the language model is exposed to a sequence of $M$ tasks denoted as $\mathbb{T} = \{\mathcal{T}^1, \ldots, \mathcal{T}^M\}$. Each task $\mathcal{T}^t$ consists of a collection of training samples $\{(x_i^t, y_i^t)\}_{i=1}^{N_t}$, where $x_i^t$ represents the input instance, and $y_i^t$ denotes its corresponding label. Assuming that the language model is parameterized by $\theta$ and the loss function is $\mathcal{L}$, the learning objective across all tasks is to minimize the generalization error:

$$\arg\min_{\theta} \sum_{t=1}^{M} \sum_{(x^t, y^t) \in \mathcal{T}^t} \mathcal{L}(f_\theta(x^t), y^t) \tag{1}$$

However, current continual learning approaches always encounter practical limitations and challenges due to their stringent constraints, which are difficult to achieve in real-life scenarios.

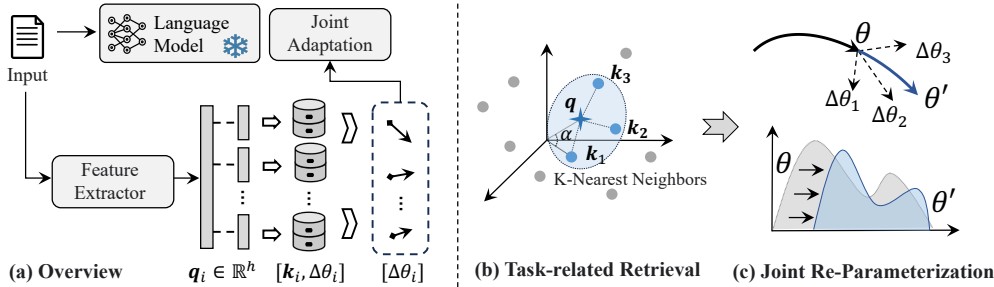

Figure 2: Illustration depicting our proposed method. $q_i, k_i \in \mathbb{R}^h$ indicate the query and key, where $h = \frac{c}{g}$ with $c$ as the channels and $g$ as the groups. The weight increment is denoted as $\Delta\theta_i$. SLM first retrieves relevant knowledge based on the task distribution and then adapts the pretrained model through joint re-parametrization to align with the corresponding task.

**Generalized continual learning.** We propose addressing this challenging problem in a more generalized setting, which effectively eliminates auxiliary operations by solely leveraging new task data, and encompasses a wider range of task types. Our goal is to achieve incremental knowledge acquisition and retention without relying on experience replay of past data, model optimization constraints, or artificial auxiliary information. Furthermore, unlike prior methods that are primarily limited to single tasks such as classification, we extend the scope of our approach to encompass diverse domains and task types within the broader spectrum of language-related tasks. This expansion allows for a more comprehensive and practical application of our proposed methodology.

## 3 SCALABLE LANGUAGE MODEL

In this study, we introduce two novel techniques, namely Joint Adaptive Re-parameterization (JARe) and Dynamic Task-related Knowledge Retrieval (DTKR), which are detailed in Sec. 3.1 and Sec. 3.2 respectively. JARe dynamically adjusts the model's weights to suit various task contexts, leveraging the knowledge priors obtained from DTKR. This adaptive mechanism enables effective scaling of the language model as illustrated in Fig. 2. Consequently, we refer to any language model that efficiently integrates and extends novel knowledge using JARe and DTKR techniques as the *Scalable Language Model* (SLM).

### 3.1 JOINT ADAPTIVE RE-PARAMETERIZATION

**Efficient tuning for continual learning.** Recent research has shown that optimizing a small subset of the model or incorporating minimal trainable parameters enables the pre-trained model to adapt to downstream tasks (Li & Liang, 2021; Houlsby et al., 2019). Based on this, recent continual learning methods have proposed to incrementally incorporate new parameters like prompts for sequential tasks while keeping the pre-trained models frozen (Razdaibiedina et al., 2023; Qin & Joty, 2021; Wang et al., 2022; Madotto et al., 2020). However, they still face certain limitations:

- Appending new parameters without pre-training may result in convergence challenges, performance degradation, and increased cost. Especially when scaling up to large language models and long prompts (Li & Liang, 2021; Hu et al., 2021), it can introduce additional training challenges.
- The new parameters are commonly stacked and accumulated together without distinguishing or relying on task-IDs before being incorporated into the model. These approaches (Razdaibiedina et al., 2023; Qin & Joty, 2021; Wang et al., 2022) still lack the capability to adaptively adjust the importance of each element based on the task distribution.

More discussions regarding the parameter-efficient tuning methods can be found in Appendix A.5.

**Joint adaptive re-parameterization.** To address these challenges, we propose an alternative model-agnostic approach called Joint Adaptive Re-parameterization (JARe), which adaptively re-parameterizes pretrained models to effectively adapt to downstream tasks based on the joint task distribution. Let $f_\theta$ represent the pretrained model, which is parametrized with the initial parameters $\theta$. The goal during fine-tuning is to adapt the language model to a specific downstream task $\mathcal{T}$

using gradient-based learning. This adaptation is guided by the following objective:

$$\arg\min_{\theta'} \sum_{(x,y)\in\mathcal{T}} \mathcal{L}_{\mathcal{T}}(f_{\theta'}(x), y), \quad \theta' = \theta + \Delta\theta, \tag{2}$$

where $\mathcal{L}_{\mathcal{T}}$ denotes the loss function specific to task $\mathcal{T}$, and $\Delta\theta$ represents the weight increment. We regard the process of assigning the corresponding weight increment from memory to fit a specific instance as the "*adaptive re-parameterization*".

Directly preserving all weight increments of the pre-trained models would result in excessive resource consumption. Therefore, following Hu et al. (2021), we only selectively update minimal weight matrices in the dense layers and leverage low-rank adaptation technique to achieve additional cost savings. Consider a specific pre-trained weight matrix of the linear layer $\boldsymbol{W}_0$. It is updated as:

$$\boldsymbol{y} = \boldsymbol{W}'\boldsymbol{x} = (\boldsymbol{W}_0 + \Delta\boldsymbol{W})\boldsymbol{x} = (\boldsymbol{W}_0 + \boldsymbol{B}\boldsymbol{A})\boldsymbol{x}, \tag{3}$$

where $\boldsymbol{W}_0 \in \mathbb{R}^{d \times k}$ is frozen, $\boldsymbol{B} \in \mathbb{R}^{d \times r}$ and $\boldsymbol{A} \in \mathbb{R}^{r \times k}$ are trainable parameters, and $r \ll \min(d, k)$. Thus each task only requires minimal trainable parameters and utilizes acceptable memory. More implementation details can be found in A.6 in the appendix.

Subsequently, we introduce the process of adaptively re-parameterizing the pre-trained models based on the joint task distribution. In the context of a specific task $\mathcal{T}^t$, the corresponding task distribution is denoted as $p_t$. Thus, after learning a sequence of tasks, a set of weight increments $\{\Delta\theta_1, ..., \Delta\theta_M\}$ is derived, where each increment is associated with one of the $M$ distributions, namely $\{p_1, ..., p_M\}$. Given a specific instance $\boldsymbol{x}$ drawn from the distribution $p$, *i.e.* $\boldsymbol{x} \sim p$, the objective is to adapt the pretrained model $f_\theta$ to the corresponding distribution, resulting in $f_\theta \rightarrow f_{\theta + \Delta\theta_p}$.

Given the discrete nature of preserved values, direct computation of precise weight increments in continuous space is infeasible. Consequently, we resort to utilizing a set of interrelated elements to approximate and estimate similar, similar to the linear interpolations used in meta-learning Triantafillou et al. (2021). To be specific, we first employ the K-nearest neighbors (KNN) algorithm to select a subset of $K$ weight increments from the most relevant distributions, denoted as $\{\Delta\theta_1, ..., \Delta\theta_K\}$, where $K \leq M$. Then, the pre-trained models are re-parametrized towards the target task as shown in Fig 2(c), which can be formulated as:

$$\theta' = \theta + \Delta\theta_p = \theta + \frac{\sum_{i=1}^{K} \mathcal{D}(p, p_i) \cdot \Delta\theta_i}{\sum_{i=1}^{K} \mathcal{D}(p, p_i)} \tag{4}$$

Here, $\mathcal{D}(\cdot)$ represents the function that measures the correlation between two distributions. In practice, we approximate the correlation by using query-key similarity distance.

**Discussion.** A single dataset can also be allocated and partitioned into multiple distributions. In practical scenarios, there are situations where the model may inadvertently retrieve unrelated or incorrect information, resulting in the erroneously selected information and worse performance. The proposed JARe effectively alleviates this issue by employing joint re-parameterization that reaches a consensus among multiple feasible directions for optimization, thus mitigating the negative impacts. Moreover, it is noteworthy that even different datasets can often share transferable knowledge. This approach leverages the shared common knowledge among closely related tasks to enhance the model's performance and improve its generalization ability.

### 3.2 DYNAMIC TASK-RELATED KNOWLEDGE RETRIEVAL

**Overview.** This section outlines the process of retrieving the most relevant knowledge. As previously mentioned, the sequentially learned knowledge can be represented as a collection of weight increments $\{\Delta\theta_1, ..., \Delta\theta_M\}$. Subsequently, each $\Delta\theta_i$ is correlated with a *key* vector $\boldsymbol{k}_i \in \mathbb{R}^c$ ($i \in 1, ..., M$), which serves to estimate the centroid of its corresponding task distribution $p_i$. This forms the *key-value* pair, *i.e.*, $[\boldsymbol{k}_i, \Delta\theta_i]$. During the inference phase, given *query* obtained from the input, the proposed Dynamic Task-related Knowledge Retrieval (DTKR) identifies the most relevant pairs based on the correlations between the *query* and *key* vectors and then re-parameterizes the pre-trained model using the corresponding *values* as Eq. 4. As for the training phase, we divide it into the *preparation stage* and the *fine-tune stage*. The *preparation stage* exclusively serves the purpose of keys generation. In the subsequent *fine-tune stage*, the keys are frozen, and the *values* are utilized for fine-tuning specific tasks, which follows the same procedure as the inference phase.

**Keys generation and knowledge retrieval.** To begin, we initialize a set of learnable parameters with a (semi) orthogonal matrix, following the methodology described in Saxe et al. (2013); Wang et al. (2022). This initialization yields a collection of initial *keys*, ensuring orthogonality between any two *keys* within the set. After that, given a tokenized input $\boldsymbol{x}$, we employ Sentence-BERT (Reimers & Gurevych, 2019), denoted as $\boldsymbol{f}_s$, to extract its semantic features. This extraction process maps the original text $\boldsymbol{x}$ to a hidden feature space, resulting in the generation of the *query* vector $\boldsymbol{q}$. Mathematically, this process can be represented as $\boldsymbol{q} = \boldsymbol{f}_s(\boldsymbol{x})$ ($\boldsymbol{x} \in \mathbb{R}^{l \times c}$, $\boldsymbol{q} \in \mathbb{R}^c$), where $l$ represents the sequence length and $c$ denotes the number of channels. It is important to note that, to maintain consistency in the mapping process during training, $\boldsymbol{f}_s$ remains frozen and unchanged.

Then, we calculate the correlations between the *query* and *keys*, and employ the $K$-nearest neighbors algorithm to retrieve the top $K$ most similar keys $\mathbb{K}_q = \{\boldsymbol{k}_1, \ldots, \boldsymbol{k}_K\}$, where $K \leq M$. The cosine similarity distance is utilized as the metric to measure the distance between the *query* and the *keys*.

During the *preparation stage*, the selected keys $\mathbb{K}_q$ undergo optimization to improve their alignment with the distribution of input instances and perform centroid estimation. The other unselected keys remain unchanged and are not affected, which can be written as:

$$\boldsymbol{k}' \leftarrow \boldsymbol{k} + \gamma \nabla_{\boldsymbol{k}} \cos(\boldsymbol{q}, \boldsymbol{k}), \quad \boldsymbol{k} \in \mathbb{K}_q, \tag{5}$$

where $\gamma$ is the learning rate and $\cos(\cdot)$ represents the cosine similarity.

However, directly utilizing such an operation for keys generation may inadvertently result in getting stuck in a local optimum, as elaborated in Appendix A.8. This occurs when only a subset of keys is constantly selected and optimized throughout the entire process, while the remaining keys are ignored and never updated. To address this problem, we propose two strategies:

- *Group-based retrieval.* Inspired by Vaswani et al. (2017), rather than retrieving directly from the entire keys set, we first partition the set into multiple equal groups. Simultaneously, the query vector $\mathbf{q} \in \mathbb{R}^c$ is also segmented into equal parts as follows:

$$\boldsymbol{q} = [\boldsymbol{q}'_1, \ldots, \boldsymbol{q}'_g], \quad \boldsymbol{q}'_i = \boldsymbol{q}_{(i-1) \cdot c/g : i \cdot c/g}, \tag{6}$$

  where $\mathbf{q}'_i \in \mathbb{R}^{c/g}$, and $g$ represents the number of groups, which is a hyperparameter. The retrieval process is conducted independently within each $\boldsymbol{q}_i$ in distinct groups, while the outcomes are subsequently aggregated across multiple groups. Group retrieval enables the model to simultaneously capture diverse patterns and relationships presented in the input data by attending to different aspects and subsets of features. Additionally, this approach enhances the robustness of the retrieval system by compensating for any potential failure of any group to capture relevant information. As a result, it facilitates a more comprehensive and expressive representation.

- *Random keys mask.* To mitigate the retriever's tendency to overly prioritize specific keys, we introduce a method called random keys mask. This technique involves randomly masking certain keys during the training process, which encourages the model to allocate attention to other elements. In practice, this objective is accomplished by randomly assigning some cosine similarity results to $-\text{Inf}$, effectively excluding them from retrieval during training. Specially,

$$\boldsymbol{S}_{i,j} = \cos(\boldsymbol{q}'_i, \boldsymbol{k}_{i,j}) \cdot (1 - \mathcal{B}(p)) - \text{Inf} \cdot \mathcal{B}(p), \tag{7}$$

  where $\mathcal{B}(p)$ represents a Bernoulli random variable that takes the value 1 with probability $p$.

Notably, only *keys* are optimized during the *preparation stage* as Eq. 5, and *values* are unchanged and still remain as the initialized learnable parameters. After the *keys* have been generated during the *preparation stage*, they are subsequently frozen and the associated *values* are adopted as adaptable weight increments to align the language models with the forthcoming tasks of continual learning. The overall training pipeline is illustrated in Algorithm 1.

## 4 EXPERIMENTS

### 4.1 EXPERIMENT SETUP

**Datasets.** We evaluate across various benchmarks with different backbones, demonstrating strong generalization capabilities. We first test our method on the widely adopted continual learning benchmarks for language models following de Masson D'Autume et al. (2019), which use five text classification datasets (Zhang et al., 2015; Chen et al., 2020) including AG News (news classification),

---

**Algorithm 1** The training pipeline of Scalable Language Model

---

    **Input:** Training sets $\{\mathcal{T}^1, \ldots, \mathcal{T}^M\}$, $\mathcal{T}^t = \{(\boldsymbol{x}_i^t, \boldsymbol{y}_i^t)\}_{i=1}^{N_t}$
    **Output:** Grouped key-value pairs $\mathcal{V}_{1,\ldots,g} = \{[\,\boldsymbol{k}, \Delta\theta\,]\}$
1: **for** $t = 1, \ldots, M$ **do**
2:     Initialize the $t$-th task's grouped key-value pairs $\mathcal{V}_{1,\ldots,g}^t$
3:     **for** $(\boldsymbol{x}_i^t, \_) \in \mathcal{T}^t$ **do**    *# The preparation stage for task $\mathcal{T}^t$*
4:         Feature extraction and Group partition $[\,\boldsymbol{q}_1', \ldots, \boldsymbol{q}_g'\,] \leftarrow \boldsymbol{q} = f_s(\boldsymbol{x}_i^t)$ via Eq. 6
5:         Calculate similarities $\boldsymbol{S}_{i,j} = \cos(\boldsymbol{q}_i', \boldsymbol{k}_{i,j}) \cdot (1 - \mathcal{B}(p)) - Inf \cdot \mathcal{B}(p)$ via Eq. 7
6:         $\mathcal{K} = \mathcal{K}_1 \cup \cdots \cup \mathcal{K}_g$, where $\mathcal{K}_j \leftarrow$ Top-$K$ similar keys of group $j$ $(j \in \{1, \ldots, g\})$
7:         Update $\boldsymbol{k}_{i,j} \in \mathcal{K}$ by $\boldsymbol{k}_{i,j} \leftarrow \boldsymbol{k}_{i,j} + \gamma \nabla_{\boldsymbol{k}_{i,j}} \cos(\boldsymbol{q}_i', \boldsymbol{k}_{i,j})$ as Eq. 5
8:     **end for**
9:     **for** $(\boldsymbol{x}_i^t, \boldsymbol{y}_i^t) \in \mathcal{T}^t$ **do**    *# The fine-tune stage for task $\mathcal{T}^t$*
10:        Retrieve most related weight increments $\{\Delta\theta_1 \ldots \Delta\theta_K\}$ with similarity distances $\mathcal{D}$
11:        Obtain the weight increment $\Delta\theta = \sum_{i=1}^{K} \mathcal{D}_i \cdot \Delta\theta_i / \sum_{i=1}^{K} \mathcal{D}_i$ used in Eq. 4
12:        Calculate sample loss $\mathcal{L}_i = \mathcal{L}(f_{\theta+\Delta\theta}(\boldsymbol{x}_i^t), \boldsymbol{y}_i^t)$
13:        Back-propagate the gradients $\Delta_\theta \mathcal{L}_i$ to update $\{\Delta\theta_1 \ldots \Delta\theta_K\}$
14:     **end for**
15:     $\mathcal{V}_{1,\ldots,g} \leftarrow \mathcal{V}_{1,\ldots,g} \cup \mathcal{V}_{1,\ldots,g}^t$
16: **end for**

---

Yelp (sentiment analysis), DBPedia (Wikipedia article classification), Amazon (sentiment analysis) and Yahoo Answers (Q&A classification).

In our experiments with BERT-base backbone (Devlin et al., 2018), we follow the approaches of IDBR and ProgPromt (Razdaibiedina et al., 2023; Huang et al., 2021) employing four different task orders from the five tasks. We adopt the full supervised continual setting, where the training set and test set are the same as MbPA++ and LAMOL (de Masson D'Autume et al., 2019; Romanov et al., 2018), consisting of 115,000 training examples and 7,600 test examples for each task. On the contrary, we conduct the few-shot continual learning setup with T5-large backbone (Raffel et al., 2020), following the approach of LFPT5 (Qin & Joty, 2021). This setup involves sampling 16 examples per class in the training and validation sets to evaluate the performance of our proposed method on limited training resources.

We further extend our method to large generation language models with LLaMA-2 backbone (Touvron et al., 2023) and introduce a new benchmark that spans multiple domains and task types. This benchmark includes three types of tasks: question answering (medical), multiple-choice examination (mmlu), and sentiment classification (finance) (Li et al., 2023; Hendrycks et al., 2021b;a). These tasks are drawn from domains such as medical, history, finance, and more. For each task, we randomly allocate 85% of the data to the training set and the remaining portion to the test set.

**Methods Compared.** In order to compare and evaluate the performance of our method, we have selected several baselines. The selected baselines include: *Fine-tune* (de Masson D'Autume et al., 2019; Wang et al., 2020), *Replay* (Razdaibiedina et al., 2023), *MBPA++* (de Masson D'Autume et al., 2019), *IDBR* (Huang et al., 2021), *LFPT5* (Qin & Joty, 2021) and *ProgPromt* (Razdaibiedina et al., 2023). Detailed descriptions of these methods can be found in A.12 in the Appendix.

## 4.2 IMPLEMENTATION DETAILS

**Backbones.** Our proposed method, Scalable Language Model (SLM), is a model-agnostic approach to continual learning that can be applied to various backbones. In our study, we specifically selected three different models: encoder-only **BERT**-base model (Devlin et al., 2018), encoder-decoder **T5**-large model (Qin & Joty, 2021), and decoder-only **LLaMA2**-7B model Touvron et al. (2023), covering various scales and architectures. To ensure consistency, we replicate all models from HuggingFace Transformers (Wolf et al., 2020) with corresponding pretrained weights.

**Configuration.** We conducted trials using the BERT and T5 backbones with 4 NVIDIA GeForce RTX 3090 GPUs. We set the batch size to 8 and the maximum sequence length to 512 for these experiments. Additionally, for experiments involving the LLaMA2-7B backbone, we utilized 4 NVIDIA A100 GPUs with a batch size of 2. To enhance training efficiency, we employed

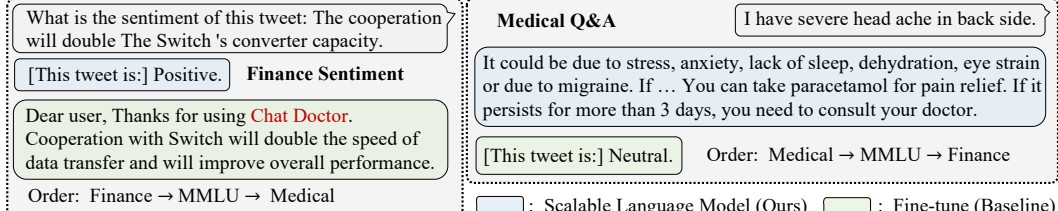

Figure 3: Comparison between our method and the baseline with the LLaMA backbone. We employ the continual training strategy to train a chat robot with diverse skills, and evaluate its performance using examples from the first task it learned. The baseline exhibits catastrophic forgetting.

DeepSpeed (Rasley et al., 2020) as a training optimization. AdamW is employed as the optimizer (Loshchilov & Hutter, 2017) for our experiments. For the *preparation stage*, we set the learning rate $lr = 1e^{-3}$ and the random mask rate $p = 20\%$ for all scenarios. Specifically, we set the learning rate to $2e^{-4}$ for fully continual learning using the BERT and LLaMA2 backbones. For the few-shot continual learning scenario with the T5 model, we set the learning rate to $2e^{-2}$. The weight decay is set to $0.01$. More configuration details can be found in Appendix A.4.

## 4.3 RESULTS ON CONTINUAL LEARNING BENCHMARKS

Table 1: Results on BERT benchmark. The results are averaged over 2 runs. "TI": whether task-ID is available during inference. "DR": whether require data replay. † and ‡ denote results from Huang et al. (2021) and Razdaibiedina et al. (2023).

| Method | TI | DR | Order 4 | 5 | 6 | 7 | Avg |
|---|---|---|---|---|---|---|---|
| Finetune† | | | 14.8 | 27.8 | 26.7 | 4.5 | 18.4 |
| Replay† | | ✓ | 67.2 | 64.7 | 64.7 | 44.6 | 57.8 |
| MBPA++† | | ✓ | 74.9 | 73.1 | 74.9 | 74.1 | 74.3 |
| IDBR† | | ✓ | 75.9 | 76.2 | 76.4 | 76.7 | 76.3 |
| **SLM** | | | **79.2** | **78.8** | **79.0** | **79.2** | **79.1** |
| ProgPrompt‡ | ✓ | | 78.0 | 77.7 | 77.9 | 77.9 | 77.9 |
| **SLM-TI** | ✓ | | - | - | - | - | **80.0** |

Table 2: Results on the continual learning with T5 backbone. All selected methods don't use **task-ID** during inference. We report the averaged results over 3 runs. † denotes results from Qin & Joty (2021).

| Method | Order 1 | 2 | 3 | Avg |
|---|---|---|---|---|
| Finetune† | 18.9 | 24.9 | 41.7 | 28.5 |
| Prompt† | 18.9 | 24.9 | 41.7 | 28.5 |
| EWC† | 39.0 | 38.0 | 44.8 | 40.6 |
| LFPT5† | 47.6 | 52.6 | 57.9 | 52.7 |
| **SLM** | **73.1** | **72.9** | **73.3** | **73.1** |

In our evaluation, we initially fine-tune the pretrained models to adapt them to sequential tasks during the training stage. Then, we assess the performance of these models on the test sets associated with each task and report the averaged scores. Experiments without the inclusion of specific notation don't provide task-ID during inference. Further, Appendix A.1 shows detailed task orders, A.3 presents the dataset details, and A.9 investigates the number of learnable parameters.

**BERT benchmark.** Tab. 1 showcases the performance of our proposed method on the BERT continual learning benchmark. Our method achieves a new state-of-the-art (SOTA) result, surpassing the alternatives, even without relying on experience replay or task-ID. Task-ID utilization simplifies the problem, particularly for methods that introduce new parameters (Razdaibiedina et al., 2023; Qin & Joty, 2021). It resembles fine-tuning on multiple tasks with distinct parameters. However, the practical determination of the input source remains challenging, such as in applications like online chatbot services where advanced knowledge of upcoming tasks may not be accessible. While our method does not depend on the task-ID, incorporating it yields a slight improvement, resulting in a remarkable performance of $80\%$ as a first achievement.

**T5 benchmark.** We conducted experiments on the few-shot continual learning benchmark for the T5 model, following Qin & Joty (2021). The results of our experiments are presented in Tab. 2, where we compare the performance of SLM with other methods. All selected methods do not require the task-ID, and only LFPT5 necessitates slight experience replay. In accordance with Qin et al. (2021) Qin & Joty (2021), we employ the text-to-text formulation for all T5 experiments, where classification labels are mapped into words. We employ accuracy as the comparative metric, considering only the first word selected as the answer from the generated output.

**LLaMA benchmark.** We extend our method to the large language model, utilizing the decoder-only LLaMA2-7B (Touvron et al., 2023) as the backbone. In our study, we incorporate three types of

tasks: question answering (medical), multiple-choice examination (mmlu), and sentiment classification (finance) across various domains. For the multiple-choice and classification tasks, we evaluate performance using accuracy. And we utilize BERTScore, following Zhang et al. (2019), to assess the medical answers generation quantity. Specially, we assign a score of 0 to the answers that do not align with the given tasks. The performance comparison with the baseline is presented in Tab. 3 and Fig. 3 provides more intuitive sampled examples. We conduct the replay methods following previous related work (He et al., 2021; Huang et al., 2021) with 1% sampled instances. It is evident that after fine-tuning sequential tasks, the baseline model has almost completely forgotten the first-learned knowledge and skills, suffering from catastrophic forgetting. And as the interval between tasks increases, the severity of forgetting tends to worsen. Indeed, our method demonstrates outstanding performance without significant forgetting. More examples can be found in Fig. 5 and Fig. 6 in the Appendix.

Table 3: Results on LLaMA benchmark. Finance: finance news sentiment classification. MMLU: multiple choice questions across multiple domains. Medical: medical question answering.

| Method | Order Finance → MMLU → Medical | | | Order Medical → MMLU → Finance | | | Avg |
|---|---|---|---|---|---|---|---|
| Finetune | 18.0 | 25.5 | 85.3 | 1.6 | 13.6 | 87.2 | 38.5 |
| Replay | 71.5 | 23.3 | 85.0 | 83.7 | 23.6 | 86.8 | 62.3 |
| **SLM** | **89.0** | **72.4** | **85.4** | **85.1** | **72.5** | **89.1** | **82.3** |

## 4.4 ANALYSIS

Table 4: The comparison of forgetting which is calculated each time after completing the training on a new task of the BERT benchmark.

| Method Order | SLM 4 | 5 | 6 | Avg | IDBR Avg |
|---|---|---|---|---|---|
| After 2 tasks | 0.0 | 0.0 | 0.0 | **0.0** | 0.8 |
| After 3 tasks | 0.0 | 0.6 | 0.4 | **0.3** | 2.4 |
| After 4 tasks | 0.2 | 0.4 | 0.8 | **0.5** | 2.7 |
| After 5 tasks | 0.5 | 0.5 | 0.5 | **0.5** | 2.9 |

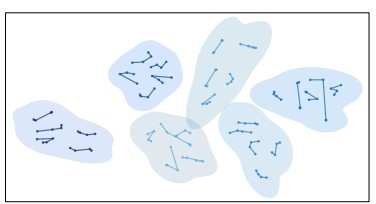

Figure 4: t-SNE visualization of keys distribution. Different spans indicate distinct groups, and the same tasks are linked by lines.

**Influence of task sequence length on forgetting.** In addition to accuracy, forgetting is another important indicator for assessing continual learning. Following the approach of Huang et al. (2021) and Chaudhry et al. (2018), we conduct experiments on the BERT benchmark and measure forgetting $\mathcal{F}_k$ after training on task $t$ using the following formula:

$$\mathcal{F}_k = \mathbb{E}_{j=1...t-1} f_j^k, \quad f_j^k = \max_{l \in \{1...,t-1\}} a_{l,j} - a_{t,j}, \tag{8}$$

where $a_{l,j}$ is the accuracy on task $j$ after trained on task $l$. We report the forgetting evaluated on each new task and report the results compared with IDBR in Tab. 4. Our method demonstrates a remarkable improvement of up to $82.8\%$ compared to the previous state-of-the-art (SOTA) approaches and all indicators are less than $0.5\%$. It effectively minimizes the forgetting of previously learned knowledge while acquiring new knowledge. Additional discussions are in Appendix A.7.

**Visualization of the keys' distribution.** To investigate the partitioning of distinct knowledge domains and assess the impact of the grouping strategy, we adopt t-SNE (Van der Maaten & Hinton, 2008) to visualize the distributions of the keys, as demonstrated in Fig. 4. In this figure, different cluster spans indicate different groups, and the keys belonging to the same task are connected by lines within each group. We can observe that different groups correspond to varied distributions, demonstrating the effectiveness of the grouping strategy in capturing diverse patterns and improving robustness. This is crucial because a single group may fail to retrieve the related information, and the presence of multiple groups helps mitigate this limitation.

**Effects of JARe.** Multiple ablation experiments were conducted to examine the impact of our proposed Joint Adaptive Re-Parameterization (JARe), and the results are presented in Tab. 5. Specifically, we replaced the weight increments in our DTKR with prompts and adapters (Li & Liang, 2021; Zhang et al., 2023). The "Separate Fine-tune" approach involves individually fine-tuning on different tasks instead of continual learning among multiple tasks. By demonstrating a marginal deviation of only **0.8%**, the proposed JARe manifests its superiority over the competitors.

Table 5: Results of the ablation studies on various storage values on BERT benchmark.

| Method | Order | | | |
| --- | --- | --- | --- | --- |
| | 4 | 5 | 6 | Avg |
| DTKR + Prompt | 54.7 | 55.8 | 49.4 | 53.3 |
| DTKR + Adapter | 71.2 | 71.2 | 70.2 | 70.9 |
| **DTKR + JARe** | **79.2** | **78.8** | **79.0** | **79.0** |
| Separate Fine-tune | - | - | - | 79.8 |

Table 6: Zero-shot evaluation on open benchmarks to assess the phenomena of forgetting and knowledge transfer.

| Method | Task | | | |
| --- | --- | --- | --- | --- |
| | Arc-c | Arc-e | Piqa | Wino |
| Finetune | 31.8 | 42.6 | 67.9 | 64.3 |
| **SLM** | **44.7** | **76.0** | 76.3 | **67.7** |
| LLaMA2 | 43.9 | 74.4 | **76.7** | 66.4 |

**Zero-shot evaluation.**   We further evaluate our method in a zero-shot setting on four open benchmarks (Arc-c, Arc-e, Piqa, Wino) (Clark et al., 2018; Sakaguchi et al., 2021; Bisk et al., 2020) following Gao et al. (2021). We first fine-tune the LLaMA-2 backbone following the order: Medical $\rightarrow$ MMLU $\rightarrow$ Finance, and then evaluate the models on the above four benchmarks. Ther results are shown in Tab. 6 and more detailed comparison can be found in A.10. It can be seen that directly utilizing fully fine-tune will result in a deterioration of the overall performance because of catastrophic forgetting. In constract to deterioriting the performance, our method even slightly improves the baseline on several tasks. It demonstrates the dual capability of our method to alleviate forgetting and effectively transfer knowledge.

## 5 RELATED WORK

**Continual Learning**, also known as lifelong learning or incremental learning, aims to improve a learning system to progressively acquire and preserve knowledge from various tasks. Existing methods for continual learning can be broadly classified into three primary categories: (1) *Replay-based* methods: periodically replay past experiences and knowledge from the observed tasks and data (Rebuffi et al., 2017; Romanov et al., 2018). The experiential data can be sampled from the previous tasks (de Masson D'Autume et al., 2019; Rebuffi et al., 2017) or synthesized using generative models (Romanov et al., 2018; Shin et al., 2017). (2) *Regularization-based* methods: impose constraints on the parameter changes of the model to prevent forgetting of previously learned tasks (Aljundi et al., 2018; Huang et al., 2021). (3) *Architecture-based* methods: employ distinct components and separate sets of parameters within the model for different tasks (Rusu et al., 2016; Mallya & Lazebnik, 2018; Razdaibiedina et al., 2023).

**Vector space model.**   Compared to traditional retrieval methods, such as the keyword-based or the rule-based, the Vector Space Model (VSM) has emerged as a prominent paradigm in information retrieval (Berry et al., 1999; Wong et al., 1987; Singhal et al., 2001). The VSM represents queries as vectors in a high-dimensional space. This representation enables the application of various similarity measures, such as cosine similarity, to determine the relevance between documents and queries (Zhang & Lu, 2003). Previous methods have endeavored to incorporate vector space retrieval into diverse endeavors (Peng et al., 2023; Danisman & Alpkocak, 2008; Wang et al., 2022), and Wang et al. (2022) adopts VSM for in-context learining. In contrast, our work introduces the use of VSM to enable dynamic transfering and adaptation of models for downstream tasks, incorporating meta-learning techniques similar to the "model soup" (Wortsman et al., 2022).

## 6 CONCLUSION

This paper presents Scalable Language Model (SLM), which enables incremental learning of sequential tasks while effectively mitigating catastrophic forgetting in a generalized setting. Notably, our approach eliminates the requirement for experience replay, optimization constraints and inference task-ID, enhancing its applicability to practical scenarios. We propose the integration of Joint Adaptive Re-Parameterization (JARe) with Dynamic Task-related Knowledge Retrieval (DTKR) to adaptively re-parameterize pretrained models based on the distance between task distributions. Our approach demonstrates remarkable stability and effectiveness across diverse model scales, leading to state-of-the-art performance on multiple benchmarks encompassing different tasks types.

The weakness of our method lies in the introduction of an additional retrieval framework, which may lead to increased computational and memory storage costs. However, when compared to the resource requirements of large models used for inference generation, this additional consumption is relatively small. Further quantitative analysis regarding this weakness can be found in Section A.11.

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

# A APPENDIX

## A.1 TASK SEQUENCE ORDERS

In standard continual learning benchmarks, the BERT and T5 models (Devlin et al., 2018; Raffel et al., 2020) utilize a total of 7 orders, as described by Huang et al. (2021); Qin & Joty (2021). The specific orders are presented in Tab. 7 as follows:

In all benchmark experiments, we initially train the pretrained model on the specific dataset, following the predefined orders mentioned above. Subsequently, we evaluate the fine-tuned model on all test sets simultaneously to test the model's performance and alleviating forgetting ability.

Table 7: Different orders of the task sequences that we used for the standard continual learning benchmarks with the BERT and T5 backbones. The 1-3 orders are used for T5 models, while the 4-7 orders are used for the BERT models.

| Order | Model | Task Sequence |
|-------|-------|---------------|
| **1** | T5 | db → amazon → yahoo → ag |
| **2** | T5 | db → amazon → ag → yahoo |
| **3** | T5 | yahoo → amazon → ag → db |
| **4** | BERT | ag → yelp → amazon → yahoo → db |
| **5** | BERT | yelp → yahoo → amazon → db → ag |
| **6** | BERT | db → yahoo → ag → amazon → yelp |
| **7** | BERT | yelp → ag → db → amazon → yahoo |

In the LLaMA benchmark, we use two orders due to resource constraints. The specific orders are listed as follows:

- Order 8: Medical → MMLU → Finance
- Order 9: Finance → MMLU → Medical

## A.2 EXAMPLES DEMO

Large Language Model (LLM) has achieved a significant success in recent years, demonstrating their distinguished ability to excel in various tasks. Furthermore, numerous applications are have been proposed that leverage fine-tuning on the pretrained large language models to adapt them to specific domains (Taori et al., 2023; Li et al., 2023). However, such operation only let the LLM grasp single domain-specific skills while potentially causing catastrophic forgetting of its general abilities.

The objective of this study is to enable the large language model (LLM) to acquire diverse skills and knowledge across multiple domains, while also possessing the potential for lifelong learning capability. More examples about the comparison of our method and the baseline, which involves direct fine-tuning of the pretrained LLM on sequence tasks, are presented in Fig. 5 and Fig. 6.

- Fig. 5: Medical → MMLU → Finance.
- Fig. 6: Finance → MMLU → Medical.

The results clearly demonstrate that while fine-tuning enables the model to acquire specific knowledge, it suffers from catastrophic forgetting, which can only answer following the formats of the last task. This is even detrimental to LLM's general abilities, which is supported by the zero-shot evaluation on the open benchmarks in Sec. 4.4. In contrast, our method significantly alleviates these issues, allowing the pretrained LLM to acquire a wide range of skills across multiple domains without experiencing catastrophic forgetting.

**Disclaimer of liability.** The examples provided is for demonstration purposes only and should not be relied upon as professional advice.

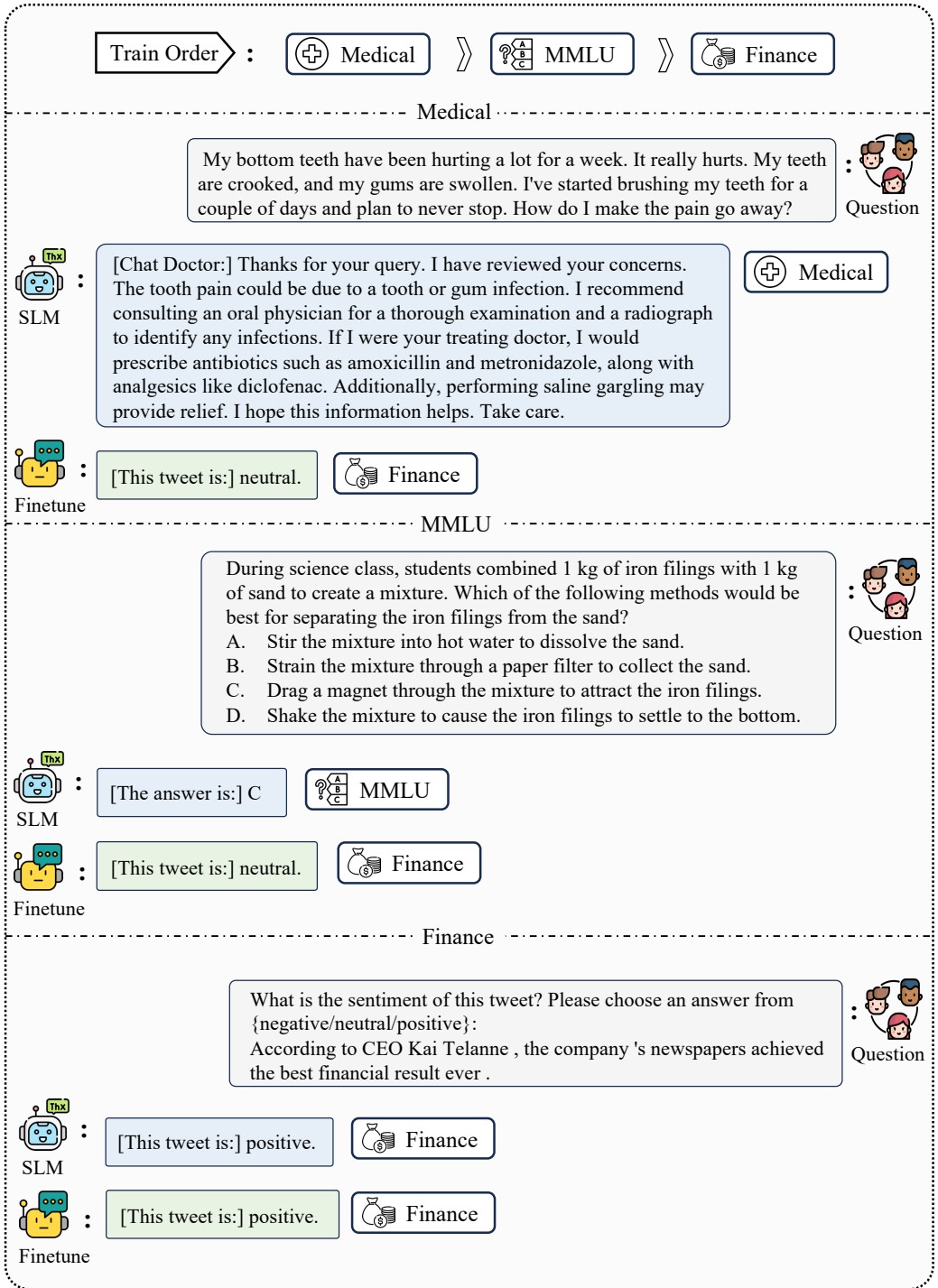

Figure 5: Demo showcases the chat robot with the LLaMA2 backbone, which undergoes continual fine-tuning on the following datasets: Medical → MMLU → Finance order.

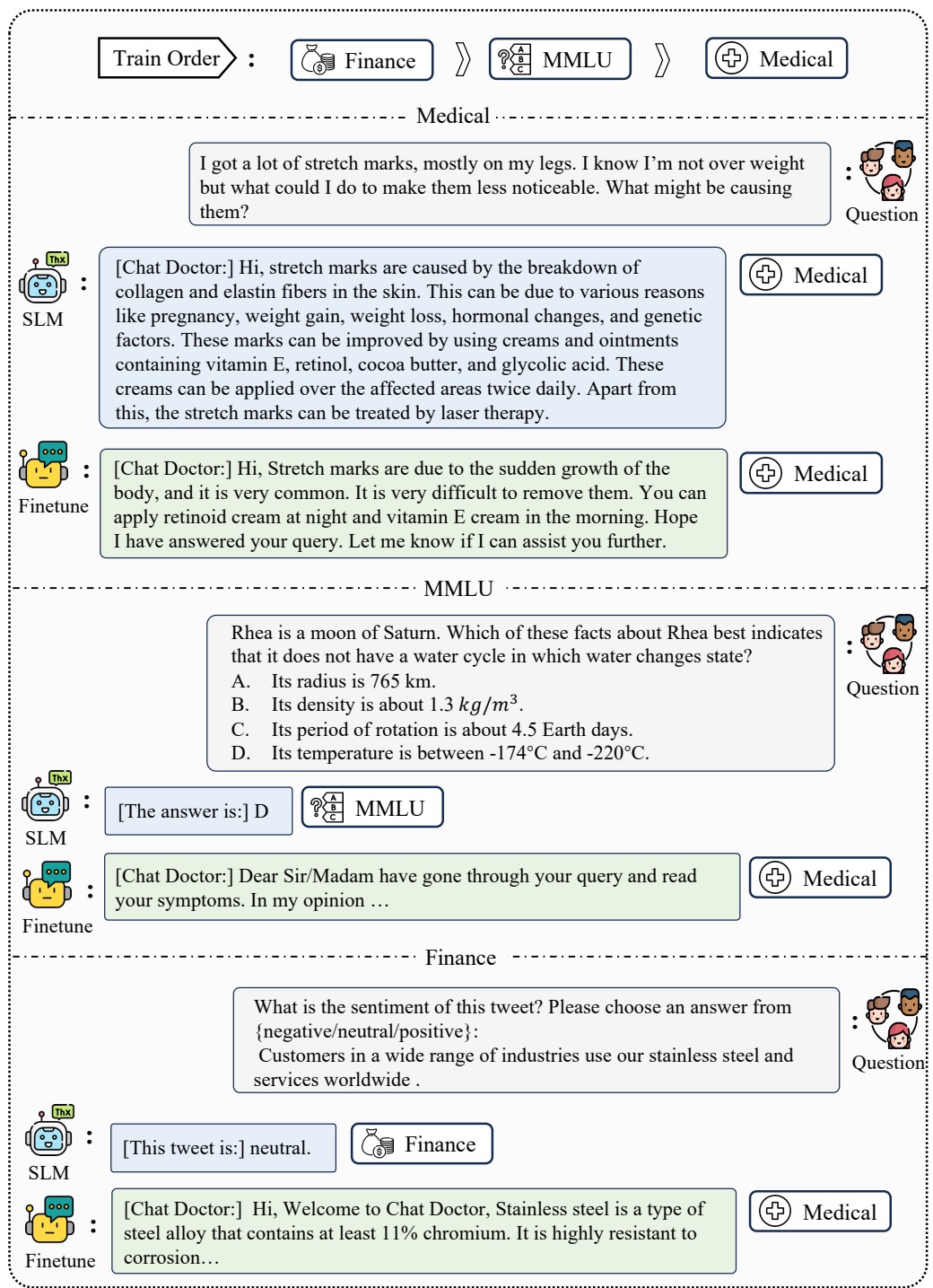

Figure 6: Demo showcases the chat robot with the LLaMA2 backbone, which undergoes continual fine-tuning on the following datasets: Finance → MMLU → Medical order.

## A.3 DATASETS

**BERT and T5 benchmarks.** More details about the five datasets used for the BERT and T5 benchmarks are listed in Tab. 8. For the BERT benchmarks, we adopt a fully supervised training approach as described in the works by Huang et al. (2021); de Masson D'Autume et al. (2019). As for the experiments conducted with T5 backbones, we use the few-shot setting where 16 examples are sampled for each class, following the methodology outlined in (Qin & Joty, 2021).

Table 8: Details of the datasets used for the BERT and T5 benchmarks. The datasets used for the BERT benchmark involve fully supervised training, while the T5 benchmark employs the few-shot setting.

| Dataset | Alias | Class | Type | Training Samples | |
| --- | --- | --- | --- | --- | --- |
| | | | | BERT | T5 |
| **AGNews** | ag | 4 | News | 115,000 | 64 |
| **Yelp** | yelp | 5 | Sentiment analysis | 115,000 | 80 |
| **Amazon** | amazon | 5 | Sentiment analysis | 115,000 | 80 |
| **DBPedia** | db | 14 | Wikipedia classification | 115,000 | 224 |
| **Yahoo** | yahoo | 10 | Yahoo Q&A | 115,000 | 160 |

**LLaMA benchmarks.** In this study, we utilize three distinct datasets(Li et al., 2023; Hendrycks et al., 2021b;a) for conducting experiments on the LLaMA benchmark. We adopt instruction tuning similar to Taori et al. (2023), while replacing the training datasets with our selected datasets. We present the sampled examples in Tab. 9 to show more details, including the instructions used in the experiments. It can be seen that in the LLaMA benchmark, the models should learn to adapt to distinct tasks across various domains with different generation formats.

Table 9: The examples of the input instances in the LLaMA benchmark.

| Dataset | Instruction | Input | Output |
| --- | --- | --- | --- |
| **Medical** | If you are a doctor, please answer the medical questions based on the patient's description. | I am suffering from bad breath , this makes me don't dare to talk and get closer to people. I can't find a way to get rid of it. | Hi, simple and effective ways to freshen your breath. Brush and floss more frequently. Scrape your tongue. Avoid foods that sour your breath. Chat Doctor. |
| **MMLU** | | Question: Since green plants make their own food, they are called? Choices: [predators, prey, decomposers, producers] | D |
| **Finance** | What is the sentiment of this tweet? Please choose an answer from {negative / neutral / positive} | According to CEO Kai Telanne , the company 's newspapers achieved the best financial result ever. | positive. |

## A.4 IMPLEMENTATION DETAILS

**Task-ID.** In this work, we mainly focus on the scenarios where inference is conducted without the task ID. In such cases, we don't know that which task or dataset is the input come from for each input instance. In particular, for architecture-based methods, having knowledge of the task ID significantly simplifies the problem by enabling direct determination of the target parameters, which

is similar to fine-tune on separate tasks independently. However, in the practical scenarios, it is always unable to determine the input task-ID directly. Such as a customer service chatbot, it doesn't have the feasibility to provide the model with the task source from the user.

**Labels.** For the T5 models, we employ a mapping technique to convert the classification labels into words, following the methodology outlined in Raffel et al. (2020). The same operation is also applied to the MMLU and Finance tasks in the LLaMA benchmark. During the evaluation phase, we select only the first word and compare it with the labels to measure accuracy, following Raffel et al. (2020). The excess part of the generation results will be ignored. Regarding the Medical task, we utilize the entire generated outputs with a maximum length of 512 and compare them with the labels using the BERTScore metric introduced by Zhang et al. (2019) following Li et al. (2023).

**Optimization hyperparameter.** AdamW (Loshchilov & Hutter, 2017) is adopted as the optimizer in all experiments. The details of the optimization hyperparameter are listed in the Tab. 10.

Table 10: The details of the optimization hyperparameter. When the number of warm-up steps is specified as a floating-point value, it represents a ratio of the total training steps.

| Benchmark | BERT | T5 | LLaMA |
|---|---|---|---|
| learning rate | $2e-4$ | $2e-2$ | $2e-4$ |
| batch size | 8 | 4 | 2 |
| epoch | 5 | 100 | 3 |
| warmup steps | 100 | 100 | 0.03 |
| weight decay | 0.01 | 0.01 | 0.01 |

### A.5 DISCUSSION OF VARIOUS PEFT METHODS

In this section, we delve into further details and compare different parameter efficient fine-tuning (PEFT) methods as the retrieved targets using the vector space retrieval framework. We replace our Joint Adaptive Re-Parameterization (JARe) with alternative components and perform ablation experiments on various individual tasks.

Prior continual learning methods have made attempts to introduce PEFT techniques like prompt tuning, which involves tuning prompts for better adaptation to new tasks. (Razdaibiedina et al., 2023; Qin & Joty, 2021; Wang et al., 2022). Specially, for a novel incremental task $\mathcal{T}_k$, the learining objective is to minimize the log probability of training examples:

$$\mathcal{L}^{lm}(\theta_{\boldsymbol{P}_k}) = - \sum_{(\boldsymbol{x},\boldsymbol{y})\in\mathcal{T}_k} \log p(\boldsymbol{y} \mid [\boldsymbol{P}_k, \boldsymbol{x}], \theta, \theta_{\boldsymbol{P}_k}), \tag{9}$$

where $\boldsymbol{P}_k$ is a learnable prompt with its corresponding parameters $\theta_k$. Similar to prompt tuning, another alternative is to replace prompts with adapters, such as prefix-tuning (Li & Liang, 2021), which also provide a flexible and modular approach to incorporate task-specific information without modifying its base parameters. We conduct the ablation experiments on the different single tasks, and the results are shown in Fig. 7.

**Adaptability.** In contrast to JARe, replacing it with prompts and adapters only allows them to be retrieved without the ability to dynamically adjust the importance and significance of each element based on the distribution distance in the vector space. As a result, all the responsibility is placed on the pretrained model itself to determine the importance of attention without the additional task distribution information, which, although lost, can be valuable for effective adaptation.

**Limited trainable parameters.** Directly introducing more learnable parameters through prompts and adapters did not lead to significant improvements and fitting abilities. As more prompts and adapters are added, the input length increases significantly. However, the rate of improvement gradually slows down (Hu et al., 2021).To solve this problem, Razdaibiedina et al. (2023) proposes to introduce a res-mlp layer, specifically,

$$\boldsymbol{P}'_k = \text{MLP}_k(\boldsymbol{P}_k) + \boldsymbol{P}_k, \tag{10}$$

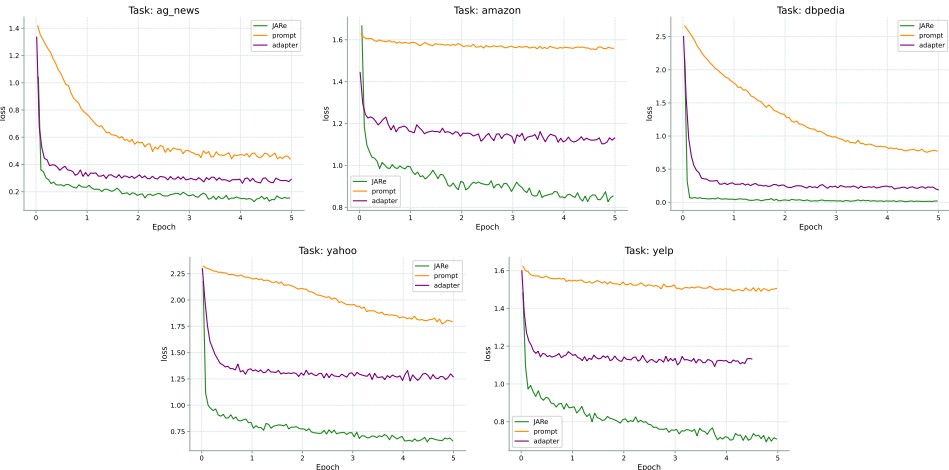

Figure 7: Comparison with different finetune methods with the vector space retrieval framework. All the experiments are conducted on a single task.

where $\text{MLP}_k(\cdot)$ is a learnable MLP layer for the task $\mathcal{T}_k$. However, the inclusion of task-ID information is inevitable when determining which MLP layer to use. But in the practical scenarios, such as a customer service chatbot, it is impossible to provide the task-ID from the users.

**Large language model.** The direct incorporation of prompt-tuning or adapters into large language models can lead to convergence difficulties and training instability. This issue primarily arises from the fact that all newly introduced parameters are randomly initialized without pretraining, which can make the model fragile when dealing with a large number of parameters. To alleviate this problem, Zhang et al. (2023) proposes to introduce a gate variable, specifically,

$$\boldsymbol{P}_i' = gate \cdot \boldsymbol{P}_i, \tag{11}$$

where $gate \in \mathbb{R}$ is a learnable parameter that is initialized as zero. This initialization ensures that the introduced parameters have no influence on the original model at first and provides a slow warm-up process. But the $gate$ variable also limits the influences of the prompts and determining its optimal value can be challenging.

In our practical experiments, we discovered an alternative approach where the introduced prompts can be initialized with tokens from the pretrained embedding layers. This initialization strategy can be effective in improving the performance and stability of the model during training. Specifically,

$$\boldsymbol{P}_i \leftarrow \gamma(\boldsymbol{E}), \tag{12}$$

where $\mathbf{E} \in \mathbb{R}^{n \times c}$ represents the pretrained embedding tokens, and $\gamma$ denotes the random selection function that returns a token randomly.

However, while these strategies may improve stability, they often do not fully overcome the upper limit bottleneck. Additionally, they can make models and inputs redundant and increase the time cost of the inference with more and more incremental tasks added.

### A.6  MODEL RE-PARAMETERIZATION

In the Sec. 3.1, we have introduce that we utilize a single group of weight increments to re-parameterize the pretrained models to adapt to a specific downstream task following Hu et al. (2021). Specifically, we freeze all the pretrained parameters without further optimization and introduce a limited number of learnable parameters to store the weight increments during training. We empoly the low-rank adaptation techniques to reduce more costs, as more details are shown in Fig. 8. Recent research further shows that optimizing all pretrained parameters for fine-tuning is unnecessary. Instead, selectively optimizing a limited set of parameters can achieve comparable performance to fully fine-tuning (Hu et al., 2021; Li & Liang, 2021; Zhang et al., 2023). Another question arises regarding which part of the pretrained models should be selected for optimization. In this work, we

Table 11: Ablation experiments results of different weight types.

| Weight | Tasks amazon | ag | yahoo | db | Average |
|--------|--------|------|-------|------|---------|
| Query | 61.1 | 94.3 | 74.7 | 99.2 | 82.3 |
| Key | 61.4 | 94.1 | 74.6 | 99.1 | 82.3 |
| Value | 62.3 | 94.3 | 75.3 | 99.2 | 82.8 |
| **Out** | **62.9** | **94.7** | **75.5** | **99.3** | **83.1** |

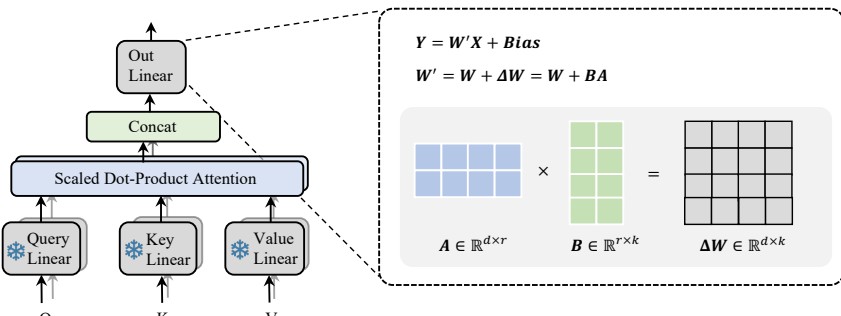

Figure 8: Illustration showcases the selection of parameter parts to store weight increments, along with the application of low-rank adaptation techniques for re-parameterizing the model.

focus on optimizing the weight matrix within the out linear layer in the attention module as shown in Fig. 8. For saving more memories, we only select a single part from the attention module, and additional ablation experiment results, showcasing the impact of different parts, are presented in Tab. 11.

## A.7 FORGETTING EVALUATION

Apart from the performance, the ability to mitigate forgetting is another crucial indicator for assessing a continual learning method. To evaluate this ability, we conduct separate tests to assess our model's performance on all previously learned tasks after training of each single incremental task. This evaluation reflects the extent to which the model retains past knowledge. The results for four different task orders on BERT benchmark are shown in Fig. 9. It is worthy to notice that as the sequence of learned tasks grows, our proposed method exhibits no significant degradation, demonstrating its remarkable ability to mitigate forgetting.

## A.8 KEYS GENERATION

**Local optimization.** In this section, we will delve into further details about the key generation process, focusing on strategies to address the issue of local optimization. When selecting and updating keys using gradient descent, it is possible that only certain keys are optimized while others are left untouched, leading to a situation of being stuck in local optimization. To tackle this issue, we propose two easy yet effective strategies: Group-based retrieval and Random keys mask. These strategies aim to capture diverse patterns and relationships within the input data by attending to different aspects and subsets of features. To evaluate the impact of these two strategies, we conducted ablation experiments, and the results are presented in Tab. 12. Specifically, with JARe, the retrieved keys are not constrained to belonging to the same task as the query, and more details we have discussed abovee. So we calculate the accuracy as follows:

$$\text{Acc} = \sum_{q \in \mathcal{T}} \delta(|\{k \in \mathbb{K}_q : k \in p(\mathcal{T})\}| > \frac{|\mathbb{K}_q|}{2}) / |\mathcal{T}|, \tag{13}$$

where $\delta(\cdot)$ is a condition function that returns 1 if the condition is satisfied. In other words, We calculate the percentage of inputs for which the retrieved keys from the same task distribution con-

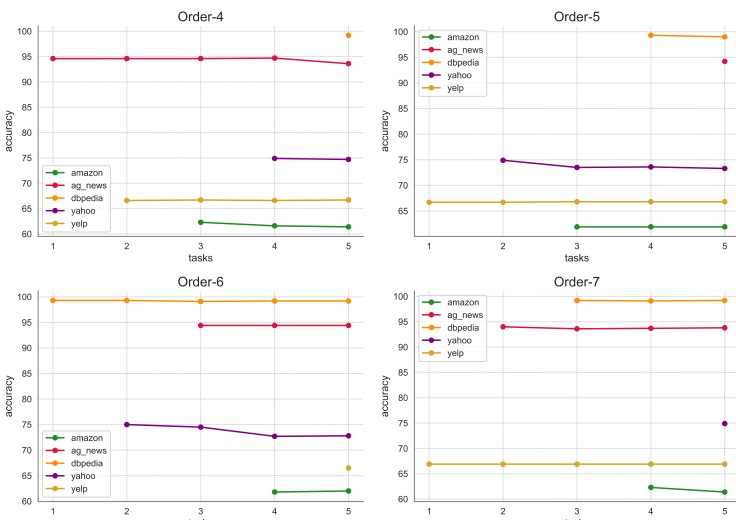

Figure 9: Accuracy of our method evaluated after training on different task sequence lengths. It is observed that as the sequence length increases, there is no obvious degradation in accuracy, indicating the significant ability of our method to alleviate forgetting.

stitute more than half of the total retrieved keys. It can be seen that our proposed strategies have significantly improved the performance, particularly in the case of the group partition operation.

Table 12: Results of the ablation experiments on our proposed keys generation strategies.

| ID | Group | Mask | Task | | | | | Avg |
|---|---|---|---|---|---|---|---|---|
| | | | ag | amazon | dbpedia | yahoo | yelp | |
| i | ✓ | ✓ | **98.0** | **98.0** | **98.6** | **86.3** | **99.5** | **96.1** |
| ii | ✓ | | 96.8 | 96.7 | 97.7 | 82.6 | 99.5 | 94.6 |
| iii | | | 91.6 | 92.2 | 91.1 | 78.9 | 93.9 | 89.5 |

**Time consumption.** For each incremental task, we split the process into two stages: (1) *Preparation stage*: generating keys for each stored value $[\boldsymbol{k}, \Delta\theta]$. In this stage, we generate keys that correspond to the stored key-value pairs. These keys play a critical role in retrieving the correct information during the subsequent fine-tuning process. (2)*Finetune stage*: fine-tune the models and preserve corresponding values. In this stage, we fine-tune the models to adapt them to the specific requirements of the downstream task. So another important consideration is the computational time required for key generation. We display the training time in Tab. 13. All experiments are conducted on $4$ NVIDIA GTX 3090 GPUs, batch_size per device is set to 4 and epoch is 3. The results show that the training time for key generation is approximately 11 minutes, which is minimal and has negligible impact on the overall process.

Table 13: Time consumption involved in generating the keys.

| Tasks | ag | amazon | dbpedia | yahoo | yelp | Avg |
|---|---|---|---|---|---|---|
| **Time(min)** | 10.2 | 8.7 | 10.5 | 14.3 | 15.4 | **11.8** |

**Similarity matrix.** To provide a visual representation of the relationships and similarities among all the generated key vectors, we present the visualization of the similarity matrix in Fig. 10. We randomly sampled some groups, where each group consists of 5 tasks and each task is associated with four keys. The visualization reveals that the keys belonging to the same task generally exhibit similar distributions, resulting in higher similarities among them. This characteristic ensures that the keys from a particular task can be easily distinguished from those of other tasks.

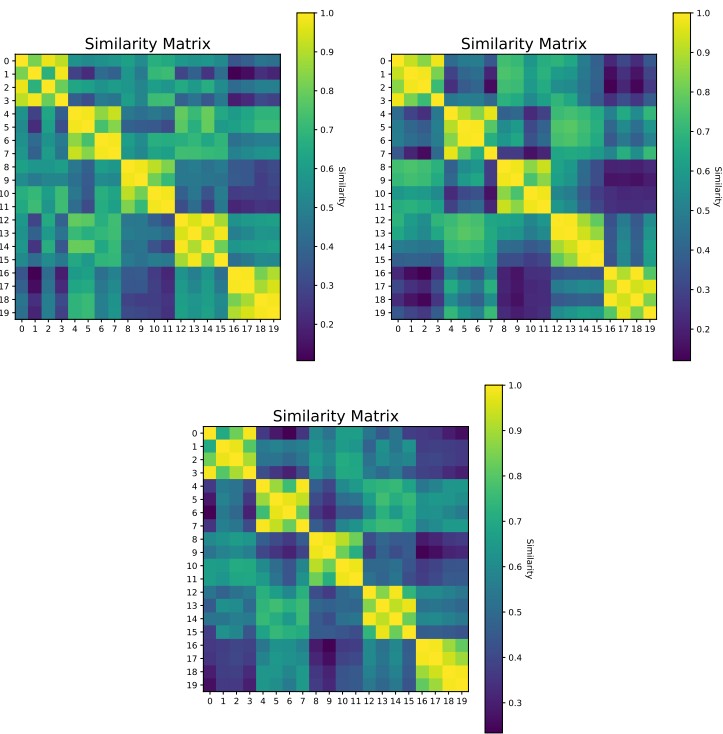

Figure 10: Visualization of similarity values between different generated key vectors.

### A.9 THE NUMBER OF LEARNABLE PARAMETERS

The size of the introduced learnable parameters does not always follow a **"less is better"** principle. Insufficient learnable parameters for certain tasks may result in underfitting, leading to unsatisfactory performance. Compared to the prompt tuning and adapters discussed in the previous sections, another advantage is its ability to dynamically adjust the size of learnable parameters within the range of 0 as the lower bound and the size of the entire model as the upper bound. This adaptability ensures its ability to dynamically adjust to diverse tasks with varying requirements.

It is important to note that while the size of the learnable parameters is a critical factor in evaluating a method, it does not have any impact on the inference process in terms of delays or computational burdens. This advantage is derived from the complete decoupling of the continual learned knowledge from the pretrained model. The cost of the retrieval process remains constant, and the retrieved values are solely utilized for model re-parameterization, without affecting the input or model scales.

To assess the impact of learnable parameter size on performance across different tasks, we conducted experiments with various hyperparameter settings, specifically modifying the size of the learnable parameters. These experiments were conducted individually for each task to evaluate the influence on performance with the BERT as the backbone.

All the experimental results are presented in Tab. 14, revealing several interesting findings and conclusions. Different tasks necessitate varying sizes of learnable parameters, and increasing the parameter size yields distinct improvements depending on the task at hand. In the case of the "yahoo" task, increasing the parameter size beyond a certain point does not provide notable benefits same as "ag" task. Our method allows us a probability to assign different sizes to the respective tasks based on their specific requirements.

Regarding the memory usage for storing these additional parameters, we provide a statistical analysis in Tab. 15. The introduced parameters in our approach are considered acceptable and relatively small compared to the size of the original pretrained large models.

Table 14: Ablation experiments to investigate the impact of learnable parameter size on performance. For all the conducted experiments, we maintained a consistent configuration of 768 channels and 12 layers for the learnable parameters.

| ID | Group | TopK | Rank | Task | | | | | Avg |
|----|-------|------|------|------|--------|---------|-------|------|-----|
| | | | | ag | amazon | dbpedia | yahoo | yelp | |
| i | 3 | 2 | 2 | 59.5 | 93.1 | 74.3 | 99.2 | 63.0 | 77.8 |
| ii | 3 | 2 | 4 | 61.0 | 94.1 | 74.8 | 99.2 | 64.2 | 78.7 |
| iii | 3 | 4 | 4 | 61.0 | 94.1 | 75.4 | 99.3 | 65.6 | 79.1 |
| iv | 6 | 2 | 8 | 62.5 | 94.4 | 75.4 | 99.4 | 66.5 | 79.7 |
| v | 6 | 2 | 12 | 63.2 | 94.5 | 75.4 | 99.3 | 67.1 | 80.0 |

Table 15: The memory consumption of the additional parameters and their proportion relative to the original model.

| | ModelSize | Additional Parameter | Proportion |
|------|-----------|----------------------|------------|
| **Bert** | 512M | 3.6M | 0.7% |
| **LLaMA** | 12.6G | 33M | 0.3% |

## A.10 ZERO-SHOT EVALUATION ON VARIOUS TASKS

Table 16: Zero-shot evaluation on open bench-marks to assess the phenomena of forgetting and knowledge transfer.

| Task | Method | Arc_e | Arc_c | Piqa | Wino |
|------|--------|-------|-------|------|------|
| Finace | Finetune | 31.8 | 42.6 | 67.9 | 64.3 |
| | **SLM** | 44.7 | 76.0 | 76.3 | 67.7 |
| MMLU | Finetune | 30.0 | 39.7 | 63.6 | 66.3 |
| | **SLM** | 49.4 | 76.7 | 76.6 | 66.2 |
| Medical | Finetune | 73.2 | 73.8 | 76.5 | 66.9 |
| | **SLM** | 44.3 | 75.0 | 77.8 | 67.8 |

Table 16 presents a more detailed zero-shot evaluation of our method using the LLaMA2 backbone finetuned on various downstream tasks. It has been observed that fine-tuning on small-scale datasets that differ significantly from the training data can have a negative impact on the LLM's generality and adaptability. Our aim is to address this issue and mitigate the catastrophic forgetting.

## A.11 WEAKNESS DISCUSSION

We humbly acknowledge that the proposed method indeed introduces a cost associated with the retrieval process. However, we find the additional cost to be acceptable because:

1. Compared to the subsequent inference models, the retrieval stage model used is notably smaller, lighter, and operates at a faster speed. This distinction is particularly significant for the T5 and Llama models.

2. In the case of generation models with the decoder architecture, each inference only produces a single token, necessitating multiple inferences to generate a complete sentence. However, the retrieval process is executed only once. Therefor, given $t_r$ as the retrivak time, $t_i$ as the inference time, $n$ as the tokens number, the proportion of time consumed is:

$$\frac{t_r}{t_r + n * t_i} * 100\%$$

We conducted an experimental comparison to measure the time consumption of different parts of various tasks using Llama on a single A100 GPU. And the results are shown in Tab. 17.

Table 17: Infference time of the retrieval framework and generalization model.

| Task | Retrieval | Generation | Sum | Proportion |
|---|---|---|---|---|
| **Finance** | 7.2ms | 147.6ms | 154.8ms | 4.7% |
| **MMLU** | 9.7ms | 161.2ms | 170.9ms | 5.7% |
| **Medical** | 8ms | 2600ms | 2608ms | 0.3% |

In terms of storage, we show the memories used for the additional parameters in Tab. 15. Moreover, it is worth noting that while our method requires additional parameters, these parameters are only used to store the weight increments. They do not incur any computational cost or increase the complexity of the original models.

## A.12   COMPARED METHODS

Below are the detailed descriptions of the methods we have chosen to compare:

- **Fine-tune** (de Masson D'Autume et al., 2019; Wang et al., 2020): Fully fine-tune all model parameters to adapt to sequential downstream tasks without additional episodic or modular components.

- **Replay** (Razdaibiedina et al., 2023): incorporates a mechanism to replay samples from previous tasks stored in the memory buffer during whole model fine-tuning, ensuring that the model retains knowledge from old tasks.

- **MBPA++** (de Masson D'Autume et al., 2019): augments the BERT model with an episodic memory module, storing **all** seen examples. It performs experience replay during training and uses K-nearest neighbors for local adaptation at test time.

- **IDBR** (Huang et al., 2021): divides the representation learning process into task-specific and task-generic spaces to attain effective representation for BERT model. This method involves continual training of the model while incorporating data replay and a regularization loss.

- **LFPT5** (Qin & Joty, 2021): leverages prompt tuning (PT) from T5 to simultaneously train the model as a task solver and a data generator. It leverages experience replaying during the learning process, requiring only a limited amount of resources.

- **ProgPromt** (Razdaibiedina et al., 2023): utilizes prompt tuning to adapt models for individual downstream tasks by employing a distinct set of prompts for each task and sequentially concatenating them with previously learned prompts. During inference, Progressive Prompts assumes that the task identifier is known, enabling the model to appropriately select the corresponding prompts.

