# OpenReview forum: "Scalable Language Model with Generalized Continual Learning"
_ICLR.cc/2024/Conference — ICLR 2024 poster_

### Official Review · Reviewer_hwoc · 2023-10-31

**Soundness:** 3 good
**Presentation:** 3 good
**Contribution:** 3 good
**Rating:** 6
**Confidence:** 2

**Summary:**

This presents Scalable Language Model (SLM), a new continual learning method for language modeling. It combines frozen large language models with Joint Adaptive ReParameterization (JARe) to adaptively adjust for new tasks. Experimental results show strong performance compared to popular continual learning methods.

**Strengths:**

1. This is first paper to utilize a large language model such as llama2-7b to study continual learning problems on a larger scale. This should bring the research in the domain to the next level.
2. Experimental results are strong.
3. The proposed method is novel, and is indeed a new paradigm for continual learning. Prior methods heavily rely on memory replay, and this method explores a very different direction.

**Weaknesses:**

1. It may be harder to reproduce the method compared to prior ones, as it is quite different from existing methods.

**Questions:**

NA

---

> ### Author Response · Authors · 2023-11-17
> **Response to Reviewer hwoc**
>
> Thank you for your valuable feedback. We are encouraged and deeply appreciate your affirmation of the performance and contribution of our work. Our responses to the mentioned weaknesses are listed below.
>
> **1."It may be harder to reproduce the method compared to prior ones, as it is quite different from existing methods."**
>
> While our approach may differ from prior methods, it is important to note that the paradigm and pipeline we propose are also straightforward and intuitive in nature. Moreover, to address the concern of reproducing our method, we will make our source code and the trained models publicly available for researchers to access and facilitate the understanding regardin the implementation details.

---

### Official Review · Reviewer_phuQ · 2023-10-31

**Soundness:** 3 good
**Presentation:** 4 excellent
**Contribution:** 2 fair
**Rating:** 6
**Confidence:** 4

**Summary:**

This paper tackles the continual learning of large language models. The major idea is to save the individual task adaptation parameters in an efficient manner by using LoRA, then during the adaptation time, they proposed a way to mix the LoRA parameters. To mix such LoRA parameters, this paper suggests defining a metric function that measures the similarity of the task by using the extracted feature from an external model, i.e., sentence BERT. The paper shows better results than prior continual learning works in LLMs in several domains.

**Strengths:**

**Strengths**

(1) The overall presentation is very good, and the writing is very clear.

(2) The method is sound and shows effective performance.

(3) This direction resembles the recent interests of model soup [1], and also some prior works in meta-learning that linearly interpolates the learned modulations to adapt the current model [2]. I think it would be great to discuss the connection.

------------

Reference\
[1] Wartsman, Model soups: averaging weights of multiple fine-tuned models improves accuracy without increasing inference time, ICML 2022\
[2] Triantafillou et al., Learning a Universal Template for Few-shot Dataset Generalization, ICML 2021

**Weaknesses:**

**Weaknesses**

(1) I disagree with the claim regarding the weakness of replay-based methods in the introduction.
- While I do agree that replay-based methods require additional storage to save the previous datasets, this paper also requires saving the parameters of the previous task's low-rank parameters and saving the keys.
- Since text dataset saving does not require much storage, **I believe comparing the storage is needed** for a fair comparison.

(2) Comparing with the upper bound performance will be interesting by assuming the ideal update is made (i.e., D(p, p_i)=0 for non-relevant tasks and D(p, p_i)=1 for the current task). Maybe such results are not an upper bound, as the method has benefits on forward transfer.

(3) Also, as the proposed and (c) task-id needed scenarios assume additional parameters (for adaptation), I believe outperforming replay-based, and regularization (e.g., EWC) is somewhat trivial. Furthermore, the proposed method requires an additional model like sentence-Bert.

(4) Due to the fact that data replay is not a big issue (as mentioned in (1)), I think adding more baselines related to data replay is needed, e.g., online SGD or other replay-based approaches [1].

------------

Overall, I quite liked the paper's approach, and the paper has the strength to be accepted to the venue. While there are weaknesses (e.g., additional parameters and network like Sentence-BERT), I believe they can be solved by re-claiming the weakness of prior methods and by highlighting the weakness of the proposed method in the manuscript, e.g., in the discussion section. Therefore, I recommend, weak acceptance, assuming that the weakness will be addressed throughout the rebuttal.

------------

**Reference**\
[1] He at al., Analyzing the Forgetting Problem in Pretrain-Finetuning of Open-domain Dialogue Response Models, EACL 2021

**Questions:**

See above

---

> ### Author Response · Authors · 2023-11-17
> **Response to Reviewer phuQ**
>
> Thank you for your insightful comments and valuable suggestions. We have incorporated more discussions with the references to provide in-depth analysis. Moreover, we will include a main paper paragraph to discuss the weaknesses of our approach. Please find our responses listed below.
>
> **1. Comparison and discussion with the replay-based methods.**
>
> Thank you for your suggestion, the discussion regarding the replay-based method is indeed crucial. The original version has included replay-method for T5 and Bert benchmarks. We then supplement the Llama experiments of the replay method follow previous work with results, which are listed below in a different order. It is worth noting that the results are better than the naive fine-tuning approach. Additionally, our method has also achieved a significant improvement, with an average accuracy of 82.3%.
>
> | Task | Finance | MMLU | Medical | Avg |
> |--------|--------|--------|--------|--------|
> |  Replay | 71.5  | 23.3 | 85.0 | 59.9 |
>
> | Task | Medical | MMLU | Finance | Avg |
> |--------|--------|--------|--------|--------|
> |  Replay | 83.7  | 23.6 | 86.9 | 64.7 |
>
> Moreover, we would like to share the following interesting findings regarding the replay-based methods.
>
> 1. The replay-based methods help maintain the LLMs ability of the format and structure of question answering. After the naive fine-tuning, the LLM can only answer in the format and knowledge related to the last task. Replay-based methods enhance the LLM's ability to maintain diverse answer formats, resulting in improvements compared to the baseline.
>
> 2. However, data replay can only preserve basic abilities like Q&A and classification. When faced with more complex tasks that demand advanced logical reasoning and inference abilities, such as MMLU, data replay can certainly give answers but may provide random and incorrect answers. This is a major drawback that contributes to the observed performance degradation of data replay on the complex task.
>
> 3. During replay, the quality of the Q&A has also deteriorated, with shorter answers that provide less useful information. The replayed methods tend to provide more generic responses and platitudes, even though the differences may not be obvious through the BertScore metric.
>
>
> **2.Require an additional model and addtitional parameters compared with the baseline**
>
> We humbly acknowledge that the proposed method indeed introduces a cost associated with the retrieval process and we are dedicated to discussing and re-claiming such weaknesses in the discussion section as you have suggested. However, we find the additional cost to be acceptable because:
>
> * Compared to the subsequent inference models, the retrieval stage model used is notably smaller, lighter, and operates at a faster speed. This distinction is particularly significant for the T5 and Llama models.
>
> * In the case of generation models with the decoder architecture, each inference only produces a single token, necessitating multiple inferences to generate a complete sentence. However, the retrieval process is executed only once. Therefor, given $t_r$ as the retrivak time, $t_i$ as the inference time, $n$ as the tokens number, the proportion of time consumed is:
> $$\frac{t_r}{t_r+n*t_i}*100\%$$
>
> We conducted an experimental comparison to measure the time consumption of different parts of various tasks using Llama on a single A100 GPU.
>
> | Task | Retrieval | Generation | All | Proportion |
> |--------|--------|--------|--------|--------|
> |  Finance | 7.2ms  | 147.6ms | 154.8ms | 4.7% |
> |  MMLU | 9.7ms  | 161.2ms | 170.9ms | 5.7% |
> | Medical  | 8ms  | 2600ms | 2608ms | 0.3% |
>
> In terms of storage, the total memory associated with storing different task datasets are list below. The "ZIP" file size represents the size of the original compressed ZIP file, while "with cache" refers to the uncompressed files with additional generated cache, such as '.arrow' files used in the HuggingFace framework. The memory cost for Language Model Models (LLMs) is also significant.
>
> | Benchmark | Bert | Llama(ZIP) | Llama(with cache) |
> |--------|--------|--------|--------|
> |  Memory | 316M  | 433M | 2.5G |
>
> Here we list the memory we used to store the additional parameters:
>
> |  | Single Task | All Task | Model Size | Proportion |
> |--------|--------|--------|--------|--------|
> |  Bert | 3.6M  | 18M | 512M | 3.5% |
> |  Llama | 33M  | 99M | 12.6G | 0.8% |
>
> Moreover, while our method requires additional parameters, these parameters are only used to store the weight increments. They do not incur any computational cost or increase the complexity of the original models.
>
>
> **3. The upper bound performance.**
>
> Thank you for your suggestions, and we are sorry for any confusion caused. Our intention was to compare the per-task fine-tuning with our methods. As you mentioned, these results cannot be considered as the upper bound and the models can also benefit from transfer learning, as demonstrated in Table 6. We will refine such statement.

---

> > ### Comment · Reviewer_phuQ · 2023-11-21
> > **Thank you for the response**
> >
> > Thank you for the response. I have read the rebuttal and am satisfied with the response. Hence, I will maintain my score. I also think the paper has the strength to be accepted to this venue.

---

> ### Author Response · Authors · 2023-11-21
> **Thank you for your thorough review**
>
> Thank you for taking the time to review our paper and for your thoughtful feedback. We greatly appreciate your careful consideration of our rebuttal and are delighted to hear that you are satisfied with our response. We sincerely appreciate your time and effort in reviewing our work. Your valuable insights have been invaluable in improving the quality of our paper.

---

### Official Review · Reviewer_US6d · 2023-10-31

**Soundness:** 3 good
**Presentation:** 3 good
**Contribution:** 4 excellent
**Rating:** 8
**Confidence:** 4

**Summary:**

- This paper proposes the Scalable Language Model (SLM) which accumulates knowledge over a sequence of tasks in a continual learning setup.
- In SLM, for each task, the task knowledge is preserved in vector space as multiple (key, value) pairs where the keys are the vector representations of the centroid of the distributions of different groups within the task distribution, and the values are the corresponding weight increments which indicate the change/delta in weight parameters over the pre-trained model parameters after training on the task dataset. Here, low-rank adaptation techniques are used to store the weight increments to avoid computational overhead.
- SLM integrates vector space retrieval into the language model to achieve scalable knowledge expansion and management
- Specifically, SLM uses (a) Dynamic Task-related Knowledge Retrieval (DTKR) to fetch the most relevant knowledge (weight increments) for an input instance from the vector space based on its distribution, and (b) Joint Adaptive Re-parameterization (JARe) to adaptively consolidate the fetched weight increments and re-parameterize the model parameters to align them with the downstream task distribution.
- In this way, SLM eliminates reliance on "regularization constraints" over weight parameters, "data replay" and "inference task-IDs", which makes it generalizable and scalable.
- Extensive experimental setup demonstrates the efficacy and stability of the model. The proposed SLM model achieves an 80% reduction in catastrophic forgetting while only losing 0.5% of the performance on the BERT benchmark.

**Strengths:**

- Originality
	- The authors have proposed a novel continual learning (CL) method, the Scalable Language Model (SLM), which eliminates the use of "regularization constraint" and "data replay" using vector space retrieval of relevant past knowledge, thus making it scalable across a variety of downstream tasks.
	- Although it seems that SLM aligns with the CL methods which incorporate additional trainable parameters for each encountered task in the sequence, however, SLM does not append additional parameters in the model itself, the knowledge acquired from the past tasks (weight increments) can be selectively utilized for future tasks thus enabling efficient knowledge transfer, and does not require the task-ID at the inference time which makes it more generalizable.
- Quality
	- The motivation is well-founded and the claims are sound.
- Clarity
	- Mathematical formulation is very detailed and explanatory.
	- Paper is clearly presented and easy to follow.
- Significance
	- The proposed SLM model is effective and consistently demonstrates state-of-the-art performance by outperforming baseline methods on diverse backbones and benchmarks.
	- Moreover, the SLM method goes beyond single task type and explores continual learning setup with diverse task types and data domains thus boosting its scalability claims.

**Weaknesses:**

- Quality
	- The number of data points in a task should also be considered while using the weight increments from that task during the retrieval step, otherwise, weight increments of the task with 100 data points will have the same importance/impact as the task with 10000 data points. Just like it happens in the weighted averaging strategy used in the FedAvg algorithm in federated learning settings.
	- The computational complexity of the proposed model seems high as the training is done for one data instance at a time and that too involves vector space retrieval of top-K relevant weight increments for each training example. Therefore, it would be fair if the authors could provide a comparison of time spent in training the proposed SLM method as compared to the baseline methods.
- Clarity
	- It would further improve the understanding of the proposed method if authors could also provide an algorithm for the inference phase.
	- It is not clear how authors have performed the initialization of (key, value) pairs for each task as mentioned in algorithm 1. It would be better if the authors could provide the mathematical formulation of this initialization strategy.
- Significance
- Typographical errors
	- [Section-3.1, Line-1] "researchs have" -> "research has" or "works have"

**Questions:**

- As the pretrained model $f_s$ is frozen and $f_{\theta}$ is always initialized with the same pre-trained initial weights, the keys of tasks with similar distributions can overlap with each other. In such cases, their values i.e., weight increments, should also be similar to each other. To understand this numerically, have authors done any such analysis to study the similarity between (key, value) pairs across tasks?
- Related to the above question, during the inference phase if DTKR provide similar (key, value) pairs from different tasks, then is there any mechanism to avoid this as they will provide redundant information and can be ignored for other relevant but non-overlapping knowledge?

---

> ### Author Response · Authors · 2023-11-17
> **Response to Reviewer US6d**
>
> Thank you for your review and valuable feedback. We appreciate your recognition of the motivation behind our work, and its simplicity and significance. Based on your feedback, we will make the necessary updates to address the mentioned typos and provide additional clarification.
>
> **1: "The number of data points in a task should also be considered..."**
>
> We sincerely appreciate your suggestion, which raises an important point regarding the consideration of the scale and size of downstream tasks. Initially, we followed the previous continual learning works, which shared the same task scale and data points. However, our approach provides flexibility and adaptability by allowing the allocation of varying numbers of data points to meet the specific requirements of each task, thereby overcoming this limitation.
>
> **2."The computational complexity of the proposed model..."**
>
> Thanks for raising this important point. We humbly acknowledge that the proposed method indeed introduces a cost associated with the retrieval process and we are dedicated to discussing and re-claiming it in the discussion section. However, we find the additional cost to be acceptable, and the benefits clearly outweight the additional minimal cost, because:
>
> * Compared to the subsequent inference models, the retrieval stage model used is notably smaller, lighter, and operates at a faster speed. This distinction is particularly significant for the T5 and Llama models.
>
> * In the case of generation models with the decoder architecture, each inference only produces a single token, necessitating multiple inferences to generate a complete sentence. However, the retrieval process is executed only once for yielding each complete answer. Therefor, given $t_r$ as the retrival time, $t_i$ as the inference time, $n$ as the tokens number, the proportion of time consumed is:
> $$\frac{t_r}{t_r+n*t_i}*100\%$$
>
> We conducted an experimental comparison to measure the time consumption of different parts of various tasks using Llama on a single A100 GPU.
>
> | Task | Retrieval | Generation | All | Per |
> |--------|--------|--------|--------|--------|
> |  Finance | 7.2ms  | 147.6ms | 154.8ms | 4.7% |
> |  MMLU | 9.7ms  | 161.2ms | 170.9ms | 5.7% |
> | Medical  | 8ms  | 2600ms | 2608ms | 0.3% |
>
>
> **3."the initialization of (key, value) pairs"**
>
> We are sorry for any confusion caused. We will provide more details about the initial process here. Please refer to [1] or the official PyTorch manual for further information. We adopt orthogonal initialization for the keys initialization, specifically,
>
> 1. Generate a random matrix $A\in R^{n\times c} \sim \mathcal{N}(0,1)$
> 2. Perform QR factorization on: $A=QR$
> 3. Adjust the sign of $Q$ based on the sign of the diagonal elements of $R$
> 4. Copy the adjusted $Q$ into the original tensor
>
> Regarding the value, we utilize zero and norm initialization.
>
> [1] Exact solutions to the nonlinear dynamics of learning in deep linear neural networks.
>
> **4. Typographical errors and improvement**
>
> Thanks for your feedback. We will address the typo and consider your suggestions for improving understanding in the final version.
>
> **5."As the pretrained model is frozen and  is always i...", "Related to the above question..."**
>
> This question is considered at the beginning of our method design. While it is inevitable to encounter unexpected results, this issue may not significantly impair the overall performance, attributed to the following facts: 1. The accuracy of the retrieval process is exceptionally high. 2. We have implemented JARe, which takes multiple results into considerations to obtain the agreed one, thereby enhancing the fault tolerance, especially in some cases where the rare outliers may occur. 3. Tasks with similar distributions may share common knowledge and characteristics. As a result, incorporating such tasks does not typically have a detrimental impact. And the similarity analysis can be found in the A.8 of the Appendix. In addition, we provide quantitative statistics about the accuracy of the retrieved related points as follows.
>
> | Task | ag_news | amazon | abpedia | yahoo | yelp |
> |--------|--------|--------|--------|--------|--------|
> |  Acc | 99.1%  | 98.1% | 99.5% | 99.2% | 97.2% |

---

### Official Review · Reviewer_3jYk · 2023-11-01

**Soundness:** 3 good
**Presentation:** 3 good
**Contribution:** 3 good
**Rating:** 6
**Confidence:** 3

**Summary:**

The authors present a novel approach aimed at continual learning of new tasks without the requirement of revisiting examples from preceding tasks, nor necessitating the taskID during the inference phase on the examples from the trained tasks. The proposed method hinges on the learning of keys and values, represented by low-rank weight matrices. Initially, the keys are learned for a task by aligning them closely with the input features derived using a frozen encoder from examples of the task. Subsequently, based on the keys, the corresponding low-rank matrices are fine-tuned. The method demonstrates superior performance compared to previous approaches.

**Strengths:**

1. The proposed method exhibits good results on continual learning benchmarks as evidenced by the experiments
2. The simplicity of the method, along with the lack of additional overhead, stands out.

**Weaknesses:**

1.The method's description appears somewhat convoluted, making it challenging to grasp the benefits fully. Relevant questions highlighting each part are in the Questions section.

2. A basic baseline employing an average feature extractor across examples of a task as a key and training a singular low-rank weight for each task as value is absent. In this method, each task would have just one key and one low-rank weight, and retrieval for examples from a new task or an existing task could be executed using either Top-1 or Top-k with existing keys.

3. The retention of performance on trained tasks seems less beneficial since for inference on an example originating from one of the trained tasks, the already trained model could simply be employed. The principal advantage of these continual systems lies in their capability to be deployed for inference as zero-shot evaluation. However, the results from Table 6 appear unconvincing as the baseline LLAMA model already exhibits decent performance. A more robust experiment could entail learning a sequence of tasks continually and evaluating the zero-shot performance with every newly learned task, please see training datasets from T0 Held-in as depicted in Figure 2 from https://arxiv.org/abs/2110.08207 for experimental setup.

**Questions:**

1. Is the preparation phase conducted collectively for all tasks, or is it executed sequentially—preparation followed by fine-tuning—for each task individually?
2. For clarification, during each task, are the keys and low-rank weights updated utilizing the examples specific to that task? Upon the introduction of a new task, are these keys and low-rank weights frozen, or are they further fine-tuned?
3. What is the initial count of keys, and is it dependent on the number of future tasks? If yes, this assumption appears unreasonable as the total count of tasks is undisclosed initially.
4. How is the aggregation across groups achieved, especially given that in each group, the query is of reduced size, and are keys of reduced size too?

---

> ### Author Response · Authors · 2023-11-17
> **Response to Reviewer 3jYk**
>
> Thank you for your valuable feedback on our paper! We appreciate your recognition of our work's performance and simplicity. We will incorporate your suggestions to refine the method's description and improve its clarity.
>
> **Q1: "Is the preparation phase conducted collectively...?", "are the keys and low-rank weights updated utilizing..."**
>
> **A1**: Following previous work, we ensure that each task is treated as an independent entity and executed sequentially. We train the corresponding keys and low-rank weights using only task-specific examples and then keep them frozen without any further fine-tuning. For your convenience, we have included a more detailed definition of the process in Algorithm 1.
>
>
> **Q2: What is the initial count of keys, and is it dependent on the number of future tasks?**
>
> **A2**: As previously mentioned, all tasks are treated as independent and decoupled, without any dependencies on the number of future tasks. The generation of task-specific keys is tailored to the fitting requirements of each task, and it is not influenced by future tasks.
>
> **Q3: How is the aggregation across groups achieved**
>
> **A3**: The aggregation across groups is accomplished through a joint weighted sum. In the retrieval process, the query is partitioned into multiple groups. Each group applies a single sub-query to multiple keys, with the keys sharing the same channels as their corresponding sub-query. This operation is similar to the multi-head attention mechanism, and more details can be found in Eq.(6) in the main paper.
>
> **1. Basic baseline**
>
> We sincerely appreciate your valuable suggestion. In response, we have incorporated the baseline results by utilizing a single key and a single weight, which are showcased in the table presented below. It is worth noting that the proposed method demonstrates a subsequent improvement in the BERT benchmark achieving a score of 79.1%. This is because relying on a single key value can result in incorrect retrieval without fault tolerance.
>
> Furthermore, we have provided a comprehensive analysis of the impact of hyperparameters, which can be found in A.8 and A.9 of the Appendix.
>
> | Order | 4 | 5 | 6 | 7 | Avg |
> |--------|--------|--------|--------|--------|--------|
> | Acc  | 77.7  | 77.9 | 77.8 | 77.5 | 77.7 |
>
> **2. Question about Table 6**
>
> Thanks for your question! We would like to clarify that:
>
> **"The retention of performance on trained tasks seems less beneficial... "**
>
> Recently, there has been a surge in fine-tuning pretrained large language models (LLMs) on domain-specific datasets to create customized assistants. However, it has been observed that fine-tuning on small-scale datasets that differ significantly from the training data can have a negative impact on the LLM's generality and adaptability, shown Table 6. Despite the performance of Llama, it exhibits significant degradation in small-scale fine-tuning tasks. And our method remidies this issue, and aims to mitigate the catastrophic forgetting and even brings about improvements in transferability.
>
> Furthermore, in the fundamental setting of continual learning, where the source of inputs is unknown and the objective is to progressively improve the model's capabilities, it is impractical to use multiple trained models and entails resource redundancy.
>
> **"zero-shot performance with every newly learned task"**
>
> Our experiments are in line with [1], where the training and zero-shot evaluation datasets cover various task types without overlap.  The results in the original main paper only include the single result of the last task, following the recent literature. We have provided the evaluation results after each individual task fine-tuning in the table below.
>
> Finance:
> | Task | Arc_c | Arc_e | Piqa | Wino |
> |--------|--------|--------|--------|--------|
> | FineTune  | 31.8 | 42.6 | 67.9 | 64.3 |
> |  SLM | 44.7 | 76.0 | 76.3 | 67.7 |
>
> MMLU:
> | Task | Arc_c | Arc_e | Piqa | Wino |
> |--------|--------|--------|--------|--------|
> | FineTune  | 30.0  | 39.7 | 63.6 | 66.3 |
> |  SLM | 49.4  | 76.7 | 76.6 | 66.2 |
>
> Meical:
> | Task | Arc_c | Arc_e | Piqa | Wino |
> |--------|--------|--------|--------|--------|
> | FineTune  | 38.9  | 67.1 | 76.5 | 66.9 |
> |  SLM | 44.3  | 75.0 | 77.8 | 67.8 |
>
> As discussed in [1], the language models (LMs) acquire transfer learning capability by learning the format and structure of question answering. However, achieving an improvement in general transfer ability necessitates large-scale multi-task training conducted collectively. This deviates from the principles of continual learning and entails significant resource requirements. These two tasks have different focuses. Our primary focus is on continual learning, ensuring fast adaptation and retaining the LLM's general abilities. Additionally, our approach shows potential for delivering fast adaptation ability to various LMs, even with small-scale training data.
>
> [1] Multitask Prompted Training Enables Zero-Shot Task Generalization

---

> > ### Comment · Reviewer_3jYk · 2023-11-19
> >
> > Thank you for clarifying the algorithm. Could you please explain how the number of keys needed for a given task is determined? I believe the number of keys would influence the final rank of an expert. Is there a method to figure this out?
> >
> > Thank you for clarifying the group aggregation
> >
> > It is encouraging to see that the proposed method performs better than the baseline. Just to clarify, did you ensure that the final number of parameters used in both methods is the same?
> >
> > Just to clarify, when a new task receives a key that has already been trained in its Top-K, is it frozen or does it continue to train? What about the corresponding Lora weight?
> >
> > It is mentioned that keys start by being orthogonal to each other. After training, are the keys made orthogonal again? How are new keys for a new task created? Are they orthogonal to each other as well as to existing keys?
> >
> > The proposed method trains keys and values corresponding to a task, assuming a source of inputs during training. Why can't we use the same knowledge for evaluating examples of tasks seen during training, instead of relying on the proposed method? I see that this method only seems beneficial for held-out datasets where we don’t have corresponding trained Lora weights. However, in Table 6, the proposed method shows no significant benefits over LLaMa2, which is just a pretrained backbone. It could be that the source tasks considered do not enable effective zero-shot generalization for the chosen datasets. This is why I recommended the T0 paper, which has both held-in dataset and held-out datasets. Could you please address this issue?
> >
> > I agree that the proposed method could potentially provide positive transfer to new tasks. Could you please direct me to the results that demonstrate this is indeed the case?

---

> ### Author Response · Authors · 2023-11-20
> **Response to Reviewer 3jYk (Part 1)**
>
> Thank you for taking the time to share your valuable feedbacks and comments. We sincerely appreciate the opportunity to provide our explanations and clarifications.
>
> **1: Explain how the number of keys needed**
>
> The selection of the number of keys is flexible and can be regarded as a hyper-parameter within the framework. However, given the fact that the consumption of each key-value pair is minimal, particularly when compared to the original model, it is acceptable to allocate an adequate number of keys at a low cost. And we found that four pairs are sufficient to ensure the desired outcome, and we found it can well generalize to different datasets and tasks.
>
> **2. The number of keys would influence the final rank**
>
> In our experiments, we discovered that the performance improvement mostly arises from the fact that multiple retrieval increase the fault tolerance mechanism and sharing knowledge. Continuously increasing parameters directly on a certain basis, such as low-rank, does not yield a substantial impact on the overall performance. More derails are shown in A.8 and A.9 in the Appendix.
>
> **3. Is the same number of parameters used in both methods?**
>
> Thanks for your kind reminder. In this experiment, we keep the number of parameters same by increasing the low-rank number. As mentioned above, the performance degradation primarily stems from incorrect retrieval and the lack of shared knowledge. It is possible that the scale of learnable parameters is already sufficient, which is also supported by recent parameter-efficient fine-tuning methods.
>
>
> **4. Is it frozen or does it continue to train**
>
> The keys are frozen after the preparation stage, indicating that they are not modified thereafter. The preparation stage is responsible for allocating and training the keys, while the fine-tuning stage focuses on adjusting the weights. Once the preparation stage is completed, the keys are frozen. More details are shown in Algorithm 1.
>
> **5. After training, are the keys made orthogonal again?**
>
> At the beginning of the preparation stage, all the allocated keys are initialized to be orthogonal to each other. Since only the top-K keys are selected and updated in each iteration, this strategy aims to make a concerted effort to retrieve different keys, enhancing the overall diversity. However, with the proposed random-mask strategy, we discovered that the initialization method has limited influence on the overall outcome. After training, all keys are adapted to the task distribution, and the initial orthogonality is no longer a constraint.
>
> **6. The source tasks considered do not enable effective zero-shot generalization for the chosen datasets**
>
> We fully endorse your point of view. The existence of similarities and shared knowledge between the source tasks and the target tasks is instrumental in significantly improving transfering effectively. Our primary objective is to showcase the exceptional performance of our methods in effectively preserving knowledge, even when applied to tasks with substantial gap in data scale, domains, and even task types.
>
> As for the benefits derived from held-in datasets, the results obtained from fine-tuning on the MMLU task can offer the evidence. Despite encompassing different domains, both the MMLU dataset and Arc datasets are multiple-choice question-answering datasets that cover various reasoning questions. They share the commonality of featuring similar question types. Our method demonstrates a remarkable improvement of up to 5.5 points, which corresponds to a relative improvement of 12.5% compared to the baseline. Notably, this improvement is achieved even without prior exposure to any questions in the ARC evaluation, highlighting the impressive capability of our methods.
>
> Finetune on MMLU:
> | Task | Arc_e | Relative Improv. | Arc_c | Relative Improv. |
> |--------|--------|--------|--------|--------|
> | SLM  | $49.4$  | $+12.5$% | $76.7$ | $+3.1$% |
> | LLama2  | 43.9 | - | 74.4 | - |
>
>
> Moreover, we also follow the previous related works to conduct the transfer learning([1], [2]). We have selected four pairs of in-pair tasks with 20-shots for T5 evaluation. Following [2], the selected pairs are: IMDb and SST2 (2-class sentiment classification), Amazon and Yelp reviews (5-class sentiment classification). And we also don't provide $\textbf{the task-ID}$ durning the transfer learning process, which makes the process more challenging, as shown below:
>
> | Task (Source→Target) | w/o transfer | w/ transfer | Relative Improv. |
> |--------|--------|--------|--------|
> | yelp → amazon  | $50.5_{±0.2}$  | $54.6_{±0.3}$ | $+8.1$%  |
> | amazon → yelp  | $51.1_{±0.6}$  | $53.9_{±1.2}$ | $+5.5$%  |
> | imdb → sst2 | $89.0_{±0.8}$  | $91.1_{±0.3}$ | $+2.3 $%  |
> | sst2 → imdb  | $90.6_{±0.6}$  | $92.4_{±0.1}$ | $+2.0$%  |
>
> [1] Pretrained transformers improve out-of-distribution robustness
> [2] Progressive prompts: continual learning for language models

---

> > ### Author Response · Authors · 2023-11-20
> > **Response to Reviewer 3jYk (Part 2)**
> >
> > **7.  Why can't we use the same knowledge for evaluating examples of tasks seen during training, instead of relying on the proposed method?**
> >
> > We would like to humbly confirm whether you are asking why we cannot train separate models for each individual task. We are glad to address this concern by considering it from three aspects:
> >
> > 1. The setting of continual learning focuses on a single model, aiming to enhance its ability for lifelong learning. This characteristic enables a model to acquire diverse skills, whereas creating individual models for each task would be resource-intensive especially for large language models (LLMs).
> >
> > 2. In real-world practical scenarios, determining the task ID or the source to which a particular task belongs can be challenging or even infeasible. For example, when developing a Q&A assistant, it may not be feasible to provide the task information by the user.
> >
> > 3. The degradation of the model's general capabilities is also a major concern. Nowadays, many NLP models utilize pretrained backbones. It may be worth considering that, while highlighting the task-specific ability, the general capabilities obtained from pretraining, as demonstrated in zero-shot evaluations, should not be overlooked.
> >
> > If there are any misunderstandings, kindly let us know, and we are open to further discussions on the matter.

---

> > > ### Comment · Reviewer_3jYk · 2023-11-21
> > >
> > > I agree with your response and thanks for your clarifications. I'd like to note that training an individual model for each task is still a form of continual learning, but maintaining such a large number of individual models can be costly. With the rise of parameter-efficient transfer learning methods like LoRA, this becomes less of an issue. We can implement a LoRA model for each new task we encounter. Even the proposed method utilizes a set of LoRA weights for every task. In such cases, we could use the LoRA model trained on a previous task for inference, instead of assuming we lack a task ID on previous tasks as the proposed approach does.
> > >
> > > Additionally, could you provide the performance for individual tasks when trained separately, to serve as a baseline for comparison? In this scenario, while there may not be positive transfer, there would also be no negative transfer. This approach still qualifies as continual learning if we consider only the tasks trained so far. I agree that it might not be effective for generalizing to new tasks, but I believe it would be a useful baseline to compare with the proposed approach.

---

> ### Author Response · Authors · 2023-11-21
> **Response to Reviewer 3jYk**
>
> Thank you for providing your valuable feedback; it has truly enriched our discussion and made it highly rewarding. We actively follow the thoughts you have mentioned to enhance both the generality and efficiency of our method. Moreover, the transfer learning capabilities we previously discussed further highlight our method's capacity to share knowledge, thus serving as a robust supplement.
>
> We apologize for any confusion caused by the separate training results. Following previous work, in the table provided, the first set of results (w/o transfer) represents the outcomes of training the tasks separately, without any influence from other tasks. Under the transfer learning setting, the reported results pertain to scenarios where the source tasks and target tasks possess similarities in terms of knowledge and task types. It is evident that our methods demonstrate the ability to share common capabilities among tasks.
>
> | Task (Source→Target) | seperately | w/ transfer | Relative Improv. |
> |--------|--------|--------|--------|
> | yelp → amazon  | $50.5_{±0.2}$  | $54.6_{±0.3}$ | $+8.1$%  |
> | amazon → yelp  | $51.1_{±0.6}$  | $53.9_{±1.2}$ | $+5.5$%  |
> | imdb → sst2 | $89.0_{±0.8}$  | $91.1_{±0.3}$ | $+2.3 $%  |
> | sst2 → imdb  | $90.6_{±0.6}$  | $92.4_{±0.1}$ | $+2.0$%  |
>
> In a more general setting of the Bert benchmark, we have conducted such experiments before. Upon reproducing these experiments with the same setting and configs, we have obtained the results of 78.9. And our method can achieve a performance of 80.0 under the same experimental conditions. It is crucial to acknowledge that the improvement achieved in this scenario may be relatively smaller compared to task-pair transfer learning, as we have previously discussed, primarily due to the larger task gap among the tasks involved.
>
> As the discussion deadline is approaching, we kindly request if there are any remaining questions or concerns, such that we can provide further responses before the deadline. Thank you once again for your valuable time.

---

### Meta-Review · Area_Chair_rh4H · 2023-12-11

**Metareview:**

This paper introduces the concept of the Scalable Language Model (SLM) designed to facilitate continual learning through a series of tasks. The design principle of SLM targets the perpetual learning of new tasks without the mandate to review examples from prior tasks or the need to supply the task ID during inference on examples retrieved from the trained tasks. The experimental results presented by the authors denote that the SLM outperforms baseline algorithms. Moreover, the implementation modality of SLM is simplistic and exhibits scalability. The overall evaluation of the paper is positive. The authors have provided diligent responses during the rebuttal phase, an effort that AC appreciates.

**Justification For Why Not Higher Score:**

The majority of the reviewers (3/4) believes that this paper has just met the threshold for acceptance (rating: 6).

**Justification For Why Not Lower Score:**

With all reviewers agreeing to its acceptance, the overall evaluation leans towards positivity.

---

### Decision · Program_Chairs · 2024-01-16

Accept (poster)